# Catalyst-free selective oxidation of C(sp$^3$)-H bonds in toluene on water

Kyoungmun Lee [1], Yumi Cho[2], Jin Chul Kim[2], Chiyoung Choi [3], Jiwon Kim[1], Jae Kyoo Lee[3,4], Sheng Li[1,5], Sang Kyu Kwak [6] ✉ & Siyoung Q. Choi [1,5] ✉

The anisotropic water interfaces provide an environment to drive various chemical reactions not seen in bulk solutions. However, catalytic reactions by the aqueous interfaces are still in their infancy, with the emphasis being on the reaction rate acceleration on water. Here, we report that the oil-water interface activates and oxidizes C(sp$^3$)-H bonds in toluene, yielding benzaldehyde with high selectivity (>99%) and conversion (>99%) under mild, catalyst-free conditions. Collision at the interface between oil-dissolved toluene and hydroxyl radicals spontaneously generated near the water-side interfaces is responsible for the unexpectedly high selectivity. Protrusion of free OH groups from interfacial water destabilizes the transition state of the OH-addition by forming π-hydrogen bonds with toluene, while the H-abstraction remains unchanged to effectively activate C(sp$^3$)-H bonds. Moreover, the exposed free OH groups form hydrogen bonds with the produced benzaldehyde, suppressing it from being overoxidized. Our investigation shows that the oil-water interface has considerable promise for chemoselective redox reactions on water without any catalysts.

Water–hydrophobe interfaces are ubiquitous in nature, including emulsions, aerosols, and colloidal suspensions. Such water interfaces provide asymmetric, heterogeneous 2D environments that function under very different laws than 3D environments we are familiar with. At the interface, the amount of water molecules forming an odd number of hydrogen bonds would be greatly enhanced[1–3], leading to the charge transfer from water to hydrophobe[4]. Moreover, an intrinsic strong electric field[5–7], high density, and alignment of molecules[8] near the aqueous interfaces could facilitate unusual chemistry in water. Those chemistry occurring at the interface of sprayed water microdroplets, which cannot be observed in bulk water, include the marked acceleration of chemical reaction rates[9,10], spontaneous redox reactions[11–16], C–N bond formation[17,18], amination[19], and carboxylation[12].

The use of water interfaces can also be expanded to include hydrophobic media, despite the fact that numerous industrial and laboratory processes do not tolerate water because of the inherent poor solubility of organic molecules and the degradation of active species in water[20]. Termed "on water" chemistry, the application of water interfaces as effective reactors for organic compounds includes Diels–Alder reactions[21], aldol reactions[22], rearrangements[23], and nucleophilic substitution[24]. These reactions can be accelerated by several hundred folds due to the stabilized transition states through the hydrogen bonds or the reagent organizations at water–hydrophobe interfaces[25,26]. An additional capability of on water chemistry is its potential to regulate regioselectivity, stereoselectivity, and chemoselectivity[21–27]. Despite the power of the water–hydrophobe interfaces to induce interesting phenomena, however, few studies have demonstrated the potential of on water chemistry to activate C(sp$^3$)–H bonds[18]. Furthermore, it is particularly rare for C(sp$^3$)–H bonds to be activated

[1]Department of Chemical and Biomolecular Engineering, Korea Advanced Institute of Science and Technology (KAIST), Daejeon, Republic of Korea. [2]School of Energy and Chemical Engineering, Ulsan National Institute of Science and Technology (UNIST), UNIST-gil 50, Ulju-gun, Ulsan, Republic of Korea. [3]Department of Applied Bioengineering, Graduate School of Convergence Science and Technology, Seoul National University, Seoul, Republic of Korea. [4]Research Institute for Convergence Science, Seoul National University, Seoul, Republic of Korea. [5]KAIST Institute for the Nanocentury, KAIST, Daejeon, Republic of Korea. [6]Department of Chemical and Biological Engineering, Korea University, Seoul, Republic of Korea. ✉e-mail: skkwak@korea.ac.kr; sqchoi@kaist.ac.kr

in highly selective manner, without the need for a catalyst, and with a significant yield.

In this work, we demonstrate that an oil–water interface effectively activates the C(sp³)–H bonds in toluene, a challenging reaction with significant industrial value[28], and oxidizes it to produce benzaldehyde with high yield under mild, catalyst-free conditions. Hydroxyl radicals (OH·) were spontaneously generated at the oil–water interface and activated the C(sp³)–H bonds in toluene to create benzaldehyde. On water reaction environment exposing free OH groups into the oil phase is responsible for exceptionally high selectivity. Toluene forms π–hydrogen bonds on water and destabilizes the OH· addition (OH-addition) transition state, whereas the hydrogen abstraction (H-abstraction) remains unchanged, thereby efficiently activating C(sp³)–H bonds to produce benzaldehyde. In addition, the produced benzaldehyde forms hydrogen bonds with the free OH groups, inhibiting its overoxidation. The obtained conversion (>99%) and selectivity (>99%) surpass previous catalytic studies conducted at high pressure or temperature[28,29] (Fig. 1). Moreover, our strategy is applicable to activate C(sp³)–H bonds and to regulate oxidation process of other aromatic compounds on water. The selective oxidation of C(sp³)–H bonds by the oil–water interface is expected to have far-reaching implications for the selective synthesis of chemicals in a green, non-toxic, and catalyst-free manner. In addition, the high electrochemical activity of reagents on water will enable the exploration of intriguing avenues for investigating the electrocatalytic potential of water–hydrophobe interfaces.

## Results

### Generation of OH· at the oil–water interfaces

We first investigated the spontaneous generation of OH· at various oil–water interfaces. The intrinsic strong electric field ($E \approx 10^9$ V/m) formed at the oil–water interface[7] is sufficient to produce OH· from hydroxide ions (OH⁻)[6,30]. We prepared three different oil–water interfaces by emulsifying 1:10, 1:1, and 10:1 (v/v) mixtures of water and hexadecane (Fig. 2a). The created emulsion systems exhibited the ratio of surface area to volume (S/V) ca. 4.3, 1.8, and 0.5 μm⁻¹ for water, and ca. 0.4, 1.8, and 4.8 μm⁻¹ for oil, respectively (Fig. 2c, see Supplementary Fig. 1 for droplet size distribution).

The formation of OH· was analyzed by hydrogen peroxide ($H_2O_2$) assay with the assumption that the generated OH· readily recombined to form $H_2O_2$[31]. We quantified the amount of $H_2O_2$ produced in water collected by centrifugation using a spectroscopic method[32] (Supplementary Fig. 2). The concentration of $H_2O_2$ increased linearly with ultrasound irradiation time, and its production rate was significantly enhanced with an increase in the S/V ratio of water (Supplementary Fig. 3a). However, the efficiency of the oil–water interfaces for OH· generation, as evaluated by $H_2O_2$ production rate per unit interfacial area, was quite comparable for all three systems (Fig. 2d). This indicates that, as previously observed[6,11,12,17–19,33–37], the spontaneous generation of OH· is a general phenomenon of water–hydrophobe interfaces. Other types of oil–water interfaces, including octane, dodecane, benzene, toluene, o-xylene, m-xylene, p-xylene, and 1,2,4-methylbenzene also produced $H_2O_2$ effectively (Supplementary Fig. 3b, c), supporting the high chemical activity of various water–hydrophobe interfaces. The decrease in $H_2O_2$ production rate induced by the shortening of the carbon chain in the oil phases may be attributed to the reduced interfacial water orientation[38], which consequently leads to a reduction in the potential interfacial area occupied by the strong electric field.

Next, we explored the utilization of the created OH· for on water chemistry. As a representative chemical reaction, the free radical polymerization of oil-soluble monomers initiated by the transport of OH· through the interfaces was examined[30] (Supplementary Fig. 4a). Dodecyl acrylate (DA) and isodecyl acrylate (IA) were tested as oil-soluble monomers. The 10:1 (v/v) mixtures of water and hexadecane solutions of DA or IA (0.4 M) were prepared and subjected to ultrasound for 2 h. Given that both DA and IA are insoluble in water, the successful polymerization of

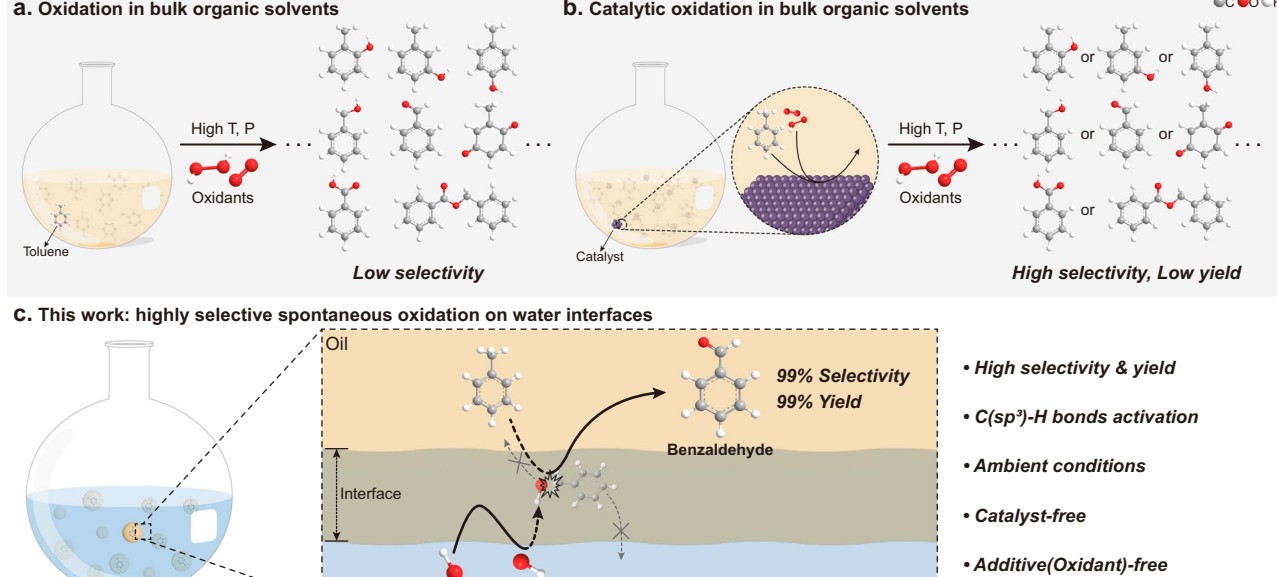

**Fig. 1 | Spontaneous and selective oxidation of toluene at the oil–water interfaces. a** Conventional synthetic procedure for toluene oxidation in bulk organic solvents. Toluene is randomly oxidized to various products at a high temperature and pressure in the presence of additional oxidants. **b** Catalytic oxidation of toluene in organic solvents. With the aid of suitable solid catalysts, toluene is selectively oxidized at a high temperature or pressure. **c** Strategy of this work for the selective oxidation of toluene on water. Under mild conditions, toluene is selectively oxidized to benzaldehyde at the oil–water interface in the absence of additional catalysts and oxidants. Spontaneously generated OH• at the oil–water interfaces effectively interacts with toluene, despite the fact that OH• and toluene is insoluble in oil and water, respectively. The distinctive 2D microenvironment at the oil–water interface, specifically the protrusion of free OH groups from interfacial water and their interaction with the toluene molecule, is responsible for the unexpectedly high selectivity for benzaldehyde.

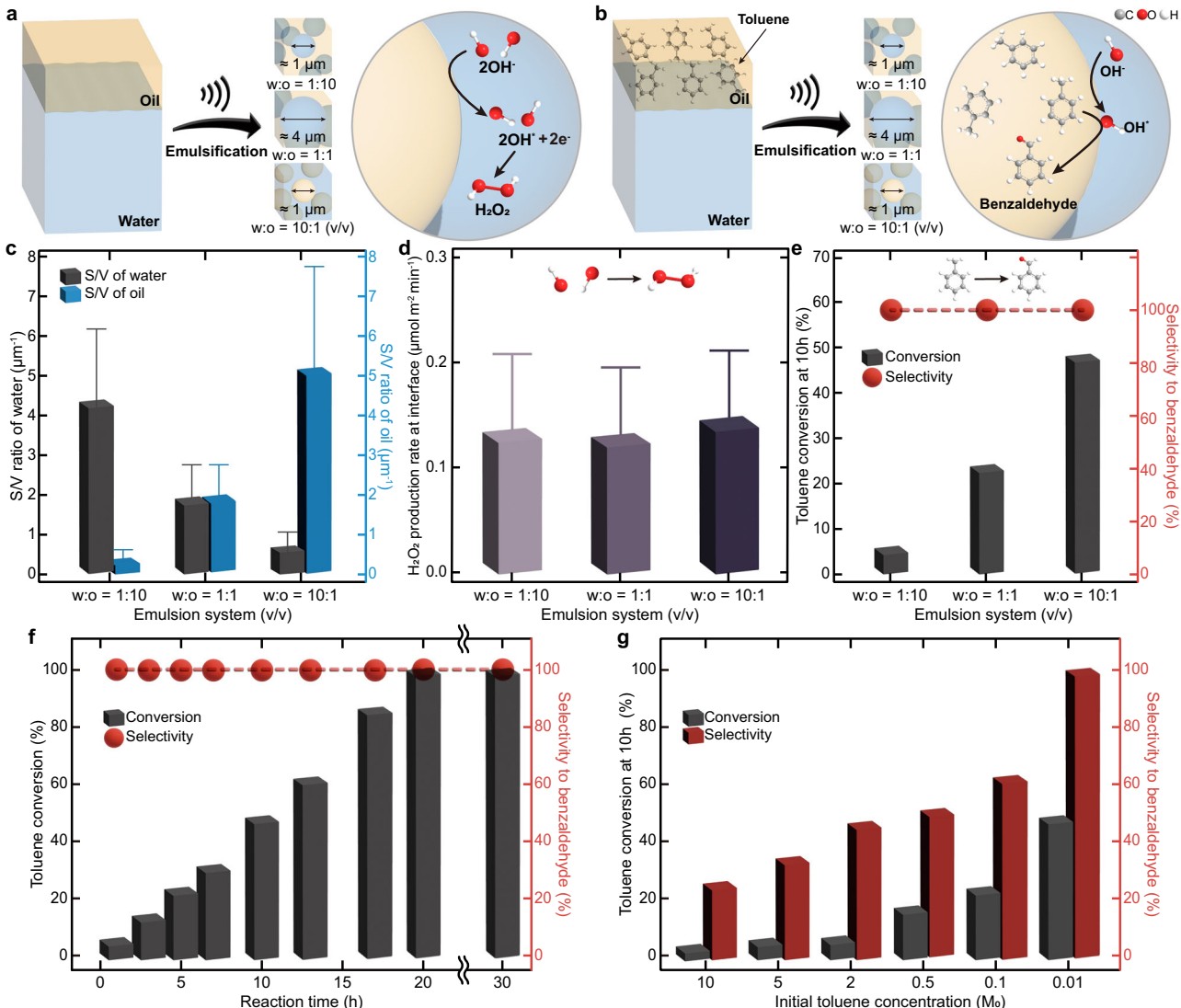

**Fig. 2 | On water chemistry at various oil–water interfaces. a** Schematic illustration of the formation of OH• and $H_2O_2$ at the oil–water interfaces. Three distinct interfaces were created by emulsifying 1:10, 1:1, and 10:1 (v/v) water and hexadecane mixtures. **b** Scheme for the oxidation of toluene at the oil–water interfaces. The produced OH• at the water-side interface reacts with toluene at the oil-side interface to selectively produce benzaldehyde. **c**–**e** S/V ratios of the water and oil phases (**c**), $H_2O_2$ production rate per unit interfacial area (**d**), and toluene conversion and selectivity for benzaldehyde at 10 h (**e**) in three different emulsion systems. Each error bar represents the standard deviation of three measurements. **f, g** The impact of reaction time (0.01 M toluene) (**f**) and initial toluene concentration (at 10 h) (**g**) on toluene conversion and selectivity to benzaldehyde: 1 atm oxygen, 25 °C (298 K), water-to-oil ratio of 10:1 (v/v).

both monomers (Supplementary Fig. 4b, c) suggests that the generated OH• could effectively interact with oil-dissolved molecules at the interface, thereby facilitating on water chemistry.

**Selective oxidation of toluene on water**

Selective oxidation of $C(sp^3)$–H bonds is of particular interest in organic synthesis as starting materials for industrial applications, such as pharmaceutical, perfume, dye, and plastics[39]. Toluene, the simplest alkyl aromatic, can be oxidized to form benzyl alcohol, benzaldehyde, and benzoic acid. These products are commercially synthesized by chlorinating toluene and then saponifying it, which requires harsh reaction conditions and generates toxic waste[40]. In addition, this procedure requires costly separation processes due to its poor selectivity. Despite the fact that numerous catalysts have been developed to enhance selectivity under mild reaction conditions[41,42], conversion efficiency exhibited a marginal improvement in order to achieve high selectivity, and the removal of catalysts is another challenging task. Inspired by the spontaneous generation of OH• at the oil–water

interfaces, which can activate C–H bonds in toluene[43], we tested their capability in the selective oxidation of toluene under mild, catalyst-free conditions utilizing the oil–water interfaces (Fig. 2b). The 1:10, 1:1, and 10:1 (v/v) mixtures of water and hexadecane solutions containing toluene (0.01 M) were prepared and irradiated with ultrasound to generate an extensive amount of oil–water interfaces. The S/V ratios of oil were ca. 0.4, 1.6, and 5.1 μm⁻¹ for water, and ca. 4.4, 1.6, and 0.5 μm⁻¹ for oil (Supplementary Fig. 5), similar to that of pure hexadecane. Encouragingly, toluene was selectively oxidized to benzaldehyde at 10 h (selectivity >99%) regardless of the different S/V ratios (Fig. 2e, Table 1). The conversion rate was proportional to the S/V ratio of oil, showing that the efficacy of the oil–water interfaces for toluene oxidation was nearly uniform. The negligible toluene oxidations in sonicated bulk solutions (Table 1, entries 9 and 10) and oil-in-water emulsions without ultrasound (Supplementary Fig. 6) support that the existence of the oil–water interfaces generated by ultrasound energy is the determinant factor for the toluene oxidation on water. Moreover, the reduced oxidation rate with increasing concentration of a radical

**Table 1 | On water activity for selective oxidation of toluene**

| Entry | w:o (v/v) | [Toluene]$_0$ (M) | Time (h) | Conv. (%)[c] | Selectivity (%)[c] | | | | |
|---|---|---|---|---|---|---|---|---|---|
| | | | | | Benzaldehyde | Benzyl alcohol | Cresol | Bibenzyl | Others[d] |
| 1 | 10:1 | 10.00 | 10 | 2.80 | 24.68 | 2.71 | 4.19 | 25.18 | 43.24 |
| 2 | 10:1 | 5.00 | 10 | 4.67 | 33.65 | 4.37 | 5.38 | 26.59 | 30.01 |
| 3 | 10:1 | 2.00 | 10 | 5.27 | 46.12 | 3.67 | 2.40 | 24.91 | 23.00 |
| 4 | 10:1 | 0.50 | 10 | 16.09 | 49.66 | 2.98 | 2.48 | 25.82 | 19.06 |
| 5 | 10:1 | 0.10 | 10 | 22.94 | 62.20 | 3.10 | 1.87 | 20.52 | 12.31 |
| 6 | 10:1 | 0.01 | 10 | 48.15 | >99 | – | – | – | – |
| 7 | 1:1 | 0.01 | 10 | 23.88 | >99 | – | – | – | – |
| 8 | 1:10 | 0.01 | 10 | 4.37 | >99 | – | – | – | – |
| 9[a] | Oil | 0.01 | 10 | – | – | – | – | – | – |
| 10[b] | MeCN | 0.01 | 10 | <0.01 | – | – | Small amount | – | – |

All reactions were conducted under 1 atm oxygen and 25 °C (298 K).
[a]Reaction was conducted in a homogeneous solution of hexadecane.
[b]Reaction was performed in a homogeneous solution of acetonitrile (MeCN, 1.9 mL) and an aqueous solution of hydrogen peroxide (35% w/w) (0.3 mL) to generate OH• by irradiating ultrasound[71].
[c]Conversion and selectivity were analyzed by gas chromatography-mass spectrometry.
[d]Other products include dimetehylbiphenyl and methyl diphenylmethane (See Supplementary Fig. 9).

scavenger 4-methoxyphenol[44], demonstrates that radical species initiate the toluene oxidation on water (Supplementary Fig. 7).

We further explored the effect of reaction time on the toluene conversion using emulsified 10:1 (v/v) mixtures of water and hexadecane solutions containing 0.01 M toluene (Fig. 2f and Supplementary Table 1). Conversion of toluene linearly increased over time and all toluene was completely oxidized after 20 h. While tiny amounts of hexadecane could be degraded to shorter alkane chains (Supplementary Fig. 8a)[35], the impact of these minor products appears to be insignificant, as evidenced by the consistent droplet size (Supplementary Fig. 8b), rate of H$_2$O$_2$ production (Supplementary Fig. 8c), and toluene oxidation rate (Fig. 2f). Regardless of the reaction time, toluene was consistently oxidized to benzaldehyde, and the generated benzaldehyde remained unchanged even after 30 h reactions. In comparison to previous catalytic studies[28,29,41,42] (Supplementary Table 2), the oil–water interface exhibited greater benzaldehyde selectivity and the capacity for full conversion under mild reaction conditions.

To estimate the oxidation mechanism at the oil–water interfaces, we compared the final products with different initial toluene concentrations (Fig. 2g). As the initial toluene concentration increased from 0.01 M to 10.00 M, both conversion and selectivity towards benzaldehyde were decreased. Moreover, various products, including benzyl alcohol, cresol, and bibenzyl emerged (Table 1 and Supplementary Fig. 9). Within the possible products, the presence of bibenzyl, which is commonly formed by the recombination of two benzyl radicals[45], implies that the benzyl radical would be an intermediate of the on-water toluene oxidation. We observed the stable C$_7$H$_7$$^+$ (m/z 91.0549), generated by the loss of one electron from the benzyl radical[14], via microdroplet mass spectrometry operating in positive mode (Supplementary Fig. 10). Reduced conversion and selectivity with decreasing dissolved oxygen concentration (Supplementary Table 3) indicate that the produced benzyl radicals seem to primarily interact with oxygen molecules and are directly oxidized to benzaldehyde[46], as opposed to forming benzyl alcohol by OH·[46,47]. Increased OH· concentration with dissolved oxygen (Supplementary Fig. 11) may assist benzaldehyde production.

Typically, toluene reaction with OH· can occur either through OH-addition or H-abstraction[46,47]. In bulk reactions, the OH-addition dominates overall reactions[43,46–48] (~90%), while the H-abstraction contributes the remaining 10%, because the OH-addition is energetically more favorable than the H-abstraction (Table 1, entry 10). In our system, however, the formation of benzyl radical and further oxidized product benzaldehyde was predominantly observed, indicating that H-abstraction is the major reaction pathway and OH· reaction with the benzene ring is suppressed at the 2D microenvironment of oil–water interfaces.

## Origin of the enhanced oxidation selectivity on water

Oil–water interfaces provide a heterogeneous microenvironment where interesting phenomena can emerge. At the oil–water interfaces, for instance, water molecules form an uneven number of hydrogen bonds and protrude free OH groups into the oil phase[1–3], which can actively participate in the formation of hydrogen bonds as hydrogen bond donors. Among the members of the hydrogen bond family is the π–hydrogen bond. Benzene rings of aromatic compounds can function as hydrogen bond acceptors to form π–hydrogen bonds. Such interactions have been well characterized in a variety of states, including the cold clusters[49], the interiors of proteins[50], liquid waters[51], and even the oil–water interfaces[52].

During the toluene oxidation at on water interfaces, where toluene reacts with OH·, the π–hydrogen bond between toluene and water may have a significant impact on reaction pathways. Toluene typically undergoes OH-addition by OH· rather than H-abstraction because the former has a lower transition state energy[46,47]. However, activation of C(sp$^3$)–H bond by H-abstraction is expected when the exposed free OH group of water forms π–hydrogen bond with the benzene ring of toluene (Fig. 3a). Toluene donates π electrons to the free OH groups of water upon the formation of the π–hydrogen bond. The lack of π electrons may destabilize the transition state of OH-addition in which the benzene ring of toluene reacts with OH·. In contrast, the deficiency of π electrons may not strongly influence on the transition state of the H-abstraction where the methyl groups of toluene react with OH· to activate C(sp$^3$)–H bonds. Thus, the benzyl radical generated by H-abstraction would rapidly react with an oxygen molecule to form benzaldehyde. Furthermore, as depicted in Fig. 3b, the synthesized benzaldehyde on water might also form hydrogen bonds with the free OH groups at the oil–water interfaces to suppress autoxidation and the formation of benzoic acid. While benzaldehyde is easily oxidized to benzoic acid on exposure to oxygen or OH·[39], it has been reported that hydrogen-bonding environments inhibit further oxidation[53].

In addition, the decrease in selectivity with increasing initial toluene concentration in Fig. 2g may point out the importance of the ratio of exposed free OH groups to aromatic compounds at the oil–water interfaces. If there are enough protruding free OH groups to form π–hydrogen bonds with the benzene ring, it could block the OH-

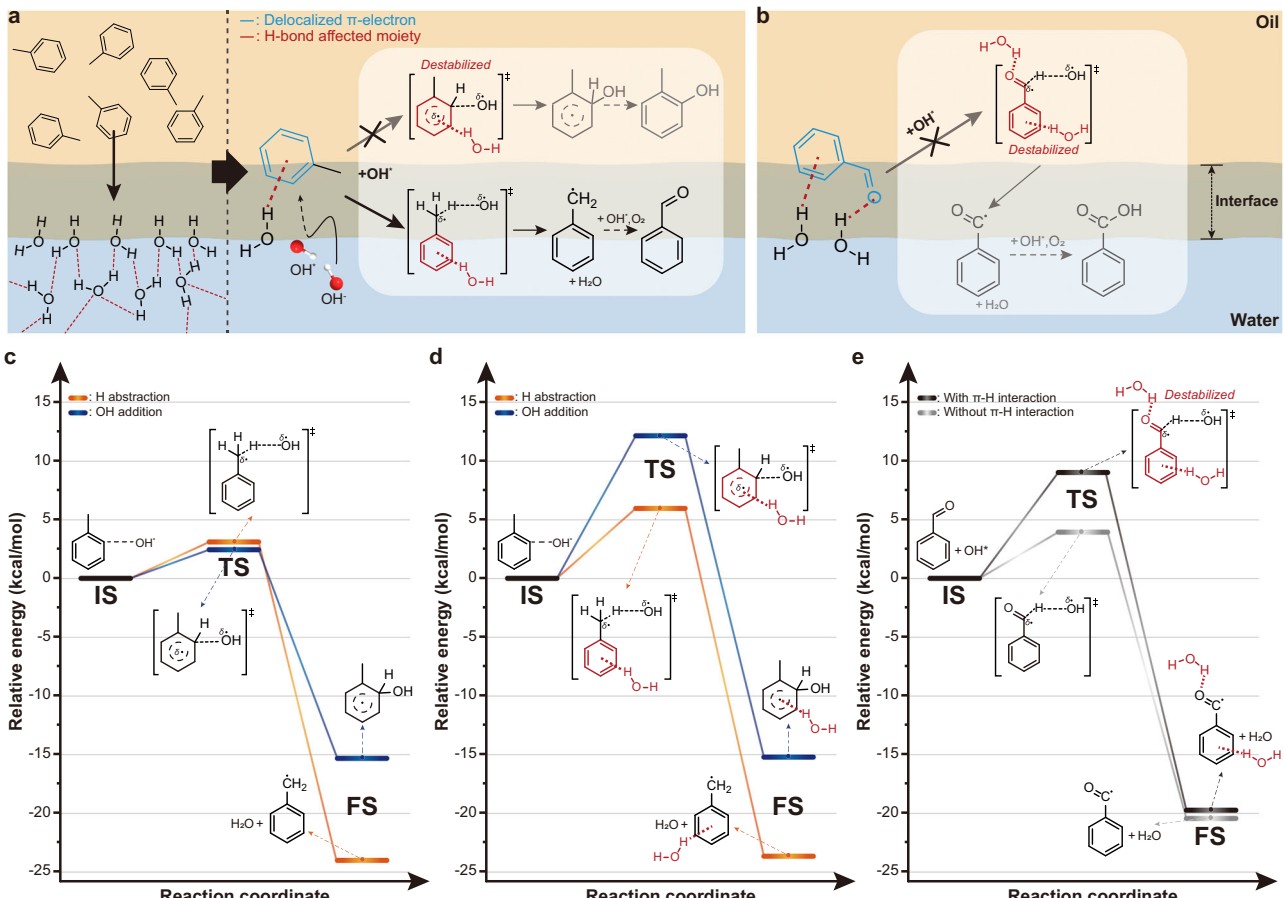

**Fig. 3 | Mechanism for the selective oxidation of C(sp³)–H bonds in toluene on water. a** Schematic showing the selective C(sp³)-H activation of toluene on water. Upon interaction with OH•, toluene forms π–hydrogen bond with the exposed free OH groups at the oil–water interface. The π–hydrogen bond destabilizes the transition state of the OH-addition, while the H-abstraction remains unchanged, resulting in the effective activation of C(sp³)–H bonds to synthesize benzaldehyde. As a representative OH-addition reaction, *ortho*-adduct was depicted. **b** Schematic illustration of benzaldehyde reaction with OH• at the phase

boundary. Hydrogen bonds formed between the produced benzaldehyde and the free OH groups on water prevent it from being further oxidized to benzoic acid. **c**, **d** Potential energy profile of toluene-OH• reaction in bulk oil (**c**), and at the oil–water interfaces (**d**). The *ortho*-addition was used to illustrate an exemplary OH-addition reaction. **e** Comparison of the enthalpy profile caused by the reaction of benzaldehyde with OH• in bulk oil and at the oil–water interfaces. IS, TS, and FS in each reaction mechanism represent the initial state, transition state, and final state, respectively.

addition and activate the C(sp³)−H bonds. However, the increased amount of toluene leads to a deficiency of π–hydrogen and the onset of the OH-addition. Nonetheless, in our system, 10 M toluene (100% v/v) still showed a selectivity higher than 90% for C(sp³)−H bond activation. Reduced selectivity for benzaldehyde may originate from increased concentration of the benzyl radical intermediates, which can be converted to benzyl alcohol by OH· and to bibenzyl by the recombination of two benzyl radicals.

To investigate the interfacial behavior of toluene at the oil–water interface, we employed density functional theory (DFT) calculations (Fig. 3c−e). We theoretically examined enthalpy profiles of H-abstraction and OH-addition in bulk oil and at the oil–water interfaces. The *ortho*-adduct reaction, which is the most energetically favorable OH-addition reaction[46,47], was selected for comparison. Note that model systems and calculation details are provided in the Simulation details section in Methods. In case of the toluene in the bulk oil (Fig. 3c and Supplementary Fig. 12), the activation energy of the H-abstraction reaction of toluene was greater than that of the OH-addition reaction. On the other hand, the intermediate formed by the H-abstraction reaction was thermodynamically favorable than that formed by the OH-addition reaction. Since there exists a crossover between activation energy and heat of reaction in reaction coordinate, the highly selective formation of benzaldehyde might not be

observable. In case of the oil–water interface (Fig. 3d and Supplementary Fig. 13), the activation energy for the OH-addition reaction of toluene significantly increased (i.e., ~9.7 kcal/mol), compared to that in the bulk oil. Also, despite a slight increase (i.e., ~2.9 kcal/mol) of the activation energy of the H-abstraction reaction, the exothermic heat of the reaction became larger in the H-abstraction reaction compared to the OH-addition reaction. Therefore, it was expected that the selective formation of benzaldehyde from the H-abstraction reaction could be higher at the oil–water interface.

To understand the different results, the interaction between toluene and water from the molecular dynamic (MD) simulation, of which configuration was brought to perform the DFT calculation, was investigated. We observed that there exist considerable π–hydrogen interactions (see the initial state in Supplementary Fig. 13a). In addition, a molecular electrostatic potential map analysis was conducted for the toluene interacting with water or oil (Supplementary Fig. 14). According to the electronic potential map, the electronic structure of toluene interacting with oil molecules showed a marginal difference from the pure toluene. However, toluene interacting with water molecules through π−hydrogen interaction showed electronic deficiency due to the electron transfer from toluene to water. Since OH· prefers electron-rich state for its attachment; it tends to act as an electron-withdrawing group when oxidizing agent is around[54], the OH·

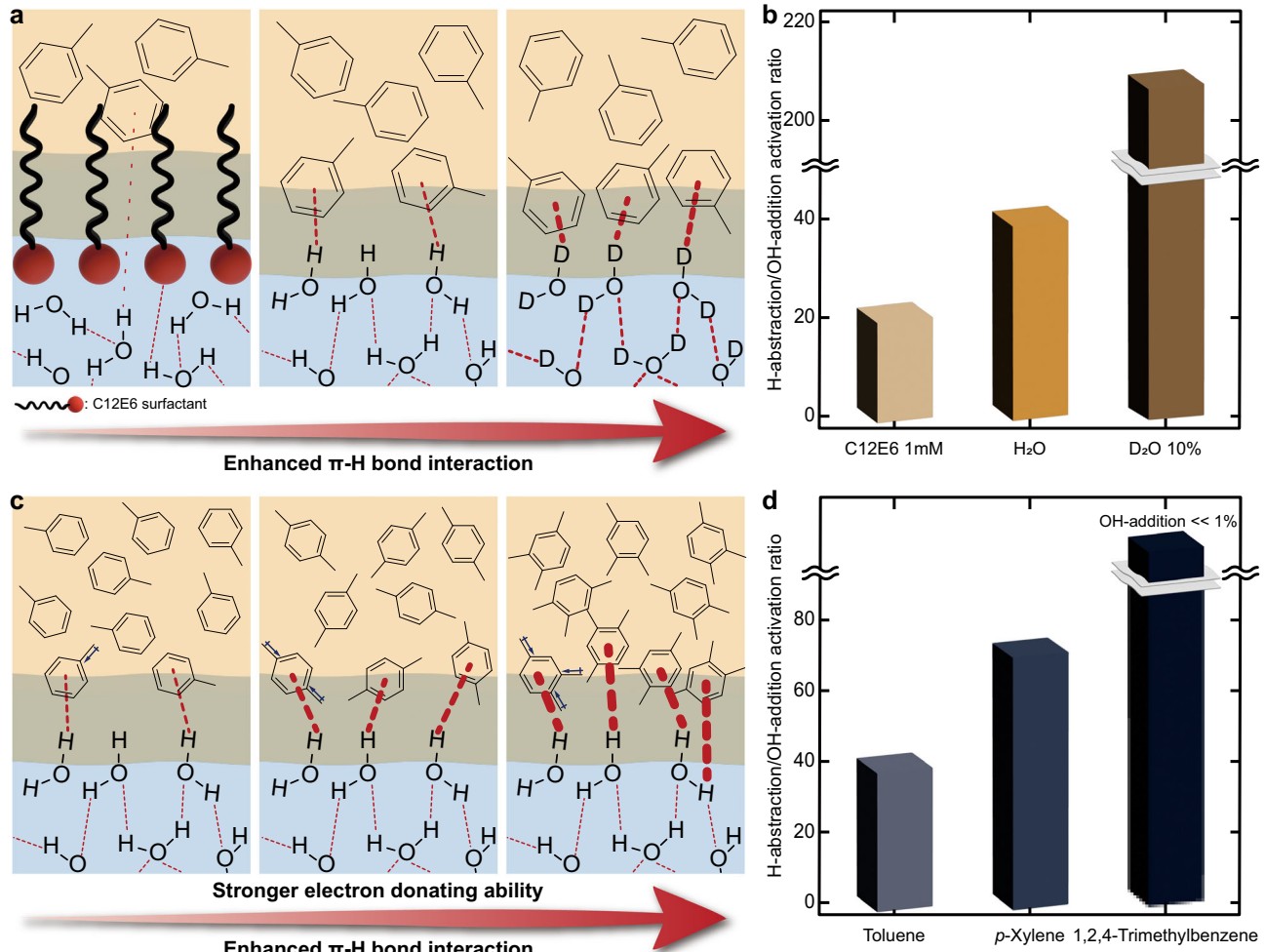

**Fig. 4 | The effect of π–hydrogen bond strength on oxidation selectivity.**
**a** Schematic depiction of the different interfacial interactions resulting from the distinct microenvironments. **b** Activation ratio of H-abstraction to OH-addition in various reaction environments. **c** Schematic showing the influence of electron-donating properties on the π–hydrogen bond strength at the oil–water interfaces. **d** Activation ratio of H-abstraction to OH-addition of three different aromatic compounds. Reaction conditions: 1 atm oxygen, 25 °C (298 K), water-to-oil ratio of 10:1 (v/v), 2 M aromatics compounds, 10 h reaction.

addition reaction becomes more challenging in the toluene interacting with water molecules. As a result, the alteration of the electronic structure driven by the π–hydrogen interaction at the oil–water interface could prevent the OH-addition reaction, allowing the selective formation of benzaldehyde. In addition, the peroxidation reaction by the H-abstraction reaction was investigated with benzaldehyde (Fig. 3e and Supplementary Fig. 15). At the oil–water interfaces, benzaldehyde exhibited two distinct forms of hydrogen bonds: π–hydrogen bonds and hydrogen bonds forming between the carbonyl group and water. It was theoretically observed that if there were no hydrogen-bonding interactions, the benzoyl radical might be formed. However, since the oil–water interfaces are majorly present, hydrogen-bonding interactions prevailed (see initial state in Supplementary Fig. 15b). Then, the H-abstraction reaction could be hindered as seen from greater activation energy (i.e., -9.0 kcal/mol) with the presence of hydrogen bonds than without it. To this end, the oil–water interface is a good environment to selectively oxidize the toluene by inducing C(sp³)-H bond activation and to prevent the peroxidation of benzaldehyde, mostly by π–hydrogen interactions.

## Impact of π–hydrogen bond strength at the oil–water interface on selective oxidation of water

To demonstrate the effect of the π-hydrogen bond on water, we further evaluated the oxidation of 2 M toluene by modifying the oil–water interfacial properties (Fig. 4a). With the addition of 1 mM hexaethylene glycol monododecyl ether (C12E6) surfactant to the water phase, the ratio of H-abstraction to OH-addition dropped from 41 to 21 (Fig. 4b, see Supplementary Fig. 16a and Supplementary Table 4 for detailed information). Since C12E6 surfactant has a strong affinity for the oil–water interfaces[55], which can efficiently impede molecular interactions across the interface[56], the declined selectivity may have been caused by the loss of π–hydrogen bonds to facilitate OH-addition. On the other hand, when 10% (v/v) of the water phase was replaced with deuterated water (D₂O), the selectivity for C(sp³)–H activation was boosted. D₂O could serve as a stronger hydrogen bond donor[57]; therefore, it may form a robust π–hydrogen bond with the benzene ring of toluene to suppress the OH-addition and strongly activate C(sp³)–H bonds.

In addition, the strength of the π-hydrogen bond was adjusted by introducing electron-donating methyl groups to the benzene ring of toluene (Fig. 4c). As the electron-donating character of substituents increases, the strength of the π-hydrogen bond could increase[58]. 2 M of toluene, p-xylene, and 1,2,4-trimethylbenzene were dissolved in hexadecane and emulsified by irradiating ultrasound to 10:1 (v/v) mixtures of water and hexadecane solutions. The ratio of H-abstraction to OH-addition improved from 41 to 74 when toluene was substituted with p-xylene (Fig. 4d, see Supplementary Fig. 16b and Supplementary Table 4 for detailed information). In addition, 1,2,4-trimethylbenzene,

which has the foremost electron-donating property and thus the most powerful π–hydrogen bonds, exhibited a dramatically increased selectivity for the C(sp³)–H activation, or H-abstraction (OH-addition <1%). Numerous correlations between π–hydrogen bonds and oxidation selectivity demonstrate that the interaction between aromatic compounds and interfacial water molecules was the crucial factor for the selective oxidation of toluene on water environments.

## Generality of selective C(sp³)–H bond activation in aromatic compounds on water interfaces

To expand our strategy for various oil–water interfaces, we first estimated toluene oxidation by varying chain lengths of hydrocarbon oils (Supplementary Table 5). Regardless of the chain length of different hydrocarbons, the generated S/V ratios were comparable to those of hexadecane, which may have been caused by the reaction system being dominated by ultrasonic energy rather than surface tension[30]. All of them showed nearly perfect selectivity for benzaldehyde production at an initial toluene concentration of 0.01 M, suggesting the significance of the 2D interfacial microenvironment irrespective of oil type. However, the conversion rate strongly decreased as hydrocarbon length decreased; this trend may be correlated with the production rate of $H_2O_2$, or OH· (Supplementary Fig. 3b).

We further evaluated the versatility of selective oxidation of various aromatic compounds, including benzene, o-xylene, m-xylene, p-xylene, and 1,2,4-methylbenzene (Supplementary Figs. 17–21 and Supplementary Table 6). For 0.01 M benzene, which does not contain an additional methyl group, nothing was produced after 10 h reaction on water. However, about 1% of benzene was converted to phenol when the oil phase was composed of 10 M benzene. This also demonstrates the crucial role of the ratio between exposed free OH groups and aromatic compounds. Similarly, 0.01 M o-xylene, m-xylene, p-xylene, and 1,2,4-methylbenzene were always oxidized to aldehyde products, whereas OH-addition was observed for 10 M o-xylene, m-xylene, p-xylene, and 1,2,4-methylbenzene.

## Discussion

We have successfully developed a synthetic strategy for activating C(sp³)–H bonds and regulating the oxidation procedure of toluene utilizing the water–hydrophobe interfaces. The large interface aqueous emulsions spontaneously generated OH·, which could be transferred through the oil–water interfaces to stimulate radical reactions with toluene. Interfacial free OH groups projecting into the oil phase interact preferentially with the benzene ring of toluene molecules to suppress the OH-addition reaction and create benzyl radical intermediate through the dominated H-abstraction, followed by oxidation to benzaldehyde using oxygen. Toluene was found to be oxidized to form benzaldehyde with high conversion (>99%) and selectivity (>99%) in 20 h under mild conditions on water without any use of added catalyst. The generated products remained unchanged even after additional exposure to the oil–water interfaces. Strong hydrogen bonds between benzaldehyde and free OH groups of interfacial water, which prevent overoxidation by destabilizing transition states, could be responsible for the high selectivity to benzaldehyde. Our strategy was also suitable for activating C(sp³)–H bonds and regulating the oxidation of other aromatic compounds on water.

As the sole oxidant, we considered OH· generated spontaneously at the oil–water interfaces. However, recent research has investigated alternative species, such as electric fields[59], water cations[59,60], and superoxide[33,59], that are produced at the interfaces. These possibilities might have a positive effect on the selective oxidation of toluene, or they might serve as minor oxidants in this study.

We suggest that the discovered sustainable synthetic methodology could minimize the conventional energy-intensive processes for the selective oxidation of various aromatic compounds. This non-

catalytic synthetic system can further reduce the amount of energy required for costly catalytic separation after final reactions. In more general terms, our findings on the utilization of the water interfaces for selective chemical synthesis may have relevant implications for the emerging field of on water chemistry. The increased chemical activity of reagents adjacent to aqueous interfaces will provide an opportunity to explore water–hydrophobe interfaces as a potential environment for facilitating interesting chemical reactions by regulating their selectivity, kinetics, and thermodynamics. Moreover, as awareness of the need for sustainable technologies grows, the role of oil–water interface as both an oxidizer and an electrocatalyst for organic molecules might be applied to a variety of fields in academia and industry.

## Methods
### Materials
Hexadecane (ReagentPlus, 99%), dodecane (RagentPlus, ≥99%), octane (puriss, p.a., ≥99.0%), benzene (anhydrous, 99.8%), toluene (anhydrous, 99.8%), p-xylene (anhydrous, ≥99%), o-xylene (anhydrous, 97%), m-xylene (anhydrous, ≥99%), 1,2,4-trimethylbenzene (98%), sodium hydroxide (NaOH, ≥97%), potassium iodine (KI, ≥99%), ammonium molybdite tetrahydrate ($(NH_4)_6Mo_7O_{24}$, ≥99.98%), potassium hydrogen phthalate ($C_8H_5KO_4$, ≥99.5%), 4-methoxylpehnol (MEHQ, RegaentPlus, 99%), aluminum oxide (activated, basic, Brockmann 1), acetonitrile (MeCN, suitable for HPLC, gradient grade, ≥99.9%), deuterium oxide ($D_2O$, 99.9 atom % D), and chloroform-d (99.8 atom % D, contains 1% (v/v) TMS) were all purchased from Sigma-Aldrich. Dodecyl acrylate (DA, stabilized with MEHQ, >98%) and isodecyl acrylate (IA, mixture of branched-chain isomers, stabilized with MEHQ) were purchased from Tokyo Chemical Industry Co., Ltd., then passed through alumina column before use to remove contained inhibitors. Hydrogen peroxide ($H_2O_2$) solution (35% w/w, Extra Pure) was purchased from Junsei Chemical Co., Ltd. Ultrapure water was produced by Millipore ICW-3000 water purification system (>18 MΩ).

### Emulsion generation
To generate large oil–water interfaces, we emulsified 1:10, 1:1, and 10:1 (v/v) mixtures of water and oil solutions by the ultrasonic bath (WUC-D06H, DAIHAN scientific), maintaining a constant water level and temperature (25–28 °C) through continuous circulations. The reaction vial (15 ml, Samwoo Science) was placed at a depth of ~7.5 cm from the transducer plate, and the reaction samples (mixtures of 200 μl water and 2000 μl oil solutions, 1100 μl water and 1100 μl oil solutions, 2000 μl water and 200 μl oil solutions in vial) were subjected to ultrasound (frequency of 40 kHz and 100% of maximum amplitude) for the desired reaction time. The size of the produced droplets was observed by optical microscopy with CCD camera (WAT-902H, Watec) and analyzed by ImageJ. The surface area to volume (S/V) ratios of water and oil were determined using the average droplet size and volume ratio of water and oil.

### Quantification of $H_2O_2$
The generation of $H_2O_2$ near the oil–water interfaces was quantified via spectroscopic method[32]. After the desired time of ultrasound irradiation, aqueous solutions were collected by centrifugation (14,100 × $g$ for 5 min, MiniSpin Plus, Eppendorf) and the $H_2O_2$ concentration was assessed. The $I^-$ ion, which has no absorption peak, is oxidized to $I_3^-$ ion having absorption peak at 353 nm[61] from the catalytic activity of ammonium molybdite under the presence of $H_2O_2$. In specific, 100 μl of two solutions, A (0.4 M KI, 0.1 M NaOH, and 0.02 mM $(NH_4)_6Mo_7O_{24}$) and B (0.1 M $C_8H_5KO_4$), were mixed with 100 μl of a diluted sample and analyzed through the absorption peak at 353 nm using UV–vis spectrometer (UV-2600, Shimadzu).

In order to examine the impact of $O_2$ on the generation of $H_2O_2$, a procedure was conducted where both water and oil were subjected to a 20-min purging process with $O_2$ gas prior to emulsification.

## Radical polymerization on water

To investigate the free radical polymerization of oil-soluble monomers on water initiated by the transport of OH· across the interfaces, desired monomer concentrations (0.4 M DA or IA) in hexadecane were prepared. Prior to irradiating the reaction mixtures, all samples were degassed with nitrogen for 20 min. The reaction samples (2000 μl water and 200 μl hexadecane solutions in a 15-ml vial) were subjected to ultrasonication (frequency of 40 kHz and 100% of maximum amplitude) for 2 h.

The number average molecular weight ($M_n$), weight average molecular weight ($M_w$), and dispersity ($M_w/M_n$) of the synthesized polymers were analyzed by size exclusion chromatography (SEC, Agilent 1260 Infinity II) with a flow rate of 1 ml/min of tetrahydrofuran (THF) as eluent at 35 °C. The SEC instrument consists of 2 × Agilent PLgel 5 μm MIXED-C columns (300 × 7.5 mm), 1260 Infinity II Quaternary pump, and 1260 Infinity II MDS refractive index detector. The system was calibrated using Agilent PS-M EasiVial calibration kit (molecular weight range: 162–364,000 g/mol). $^1$H nuclear magnetic resonance (NMR) spectra were recorded using 400 MHz Bruker Avance III HD (9.4 T) operating at ambient temperature in the chloroform-d solvent.

## Oxidation of aromatic compounds on water

Various oils with the desired concentrations of aromatic compounds were prepared and the reaction samples purged for 20 min with oxygen, unless otherwise specified. For the analysis of the radical scavenger effect on toluene oxidation, the appropriate amount of 4-methoxyphenol was additionally added to water. The oxidation reaction was conducted by subjecting the reaction samples to ultrasound (frequency of 40 kHz and 100% of maximum amplitude) for the desired reaction time.

The conversion and selectivity of aromatic compounds on water were analyzed by gas chromatography-mass spectrometry (GC-MS). The GC measurements were conducted on gas chromatograph Trace1310 (ThermoFisher) using a capillary column (TG-5MS, 30 m × 0.25 mm × 0.25 μm). The inlet temperature was maintained at 300 °C and helium was utilized as a carrier gas. Injections were carried out in split mode (split ratio 20:1, flow rate 1 ml/min, and sample volume 1 μl) at an oven temperature of 45 °C for 2 min, followed by a temperature ramp from 45 °C to 280 °C at a rate of 10 °C/min. The MS was performed on an ISQ QD300 (ThermoFisher) at a constant MS transfer line and ion source temperature of 280 °C. After solvent delay time 1.6 min, the product mass was analyzed by EI ionization mode (mass range 20–1000 amu). Phenolic byproducts (including cresol) are categorized as OH-addition contributors in the H-abstraction/OH-addition activation ratio calculation, whereas other substances are categorized as H-abstraction contributors.

## Microdroplet mass spectrometry

Toluene-in-water microdroplets are produced by injecting water (5 μl/min) and toluene (0.1 μl/min) into a spraying device from two fused silica capillaries (100.6 μm i.d., 236.2 μm o.d.). Mass spectrometry analysis was conducted at +8 kV applied voltage and 275 °C capillary temperature. Microdroplets were sprayed directly into a mass spectrometer (MS, Q-Exactive Orbitrap MS, Thermo Scientific) at 5 mm distance between the spray and the MS inlet using $N_2$ nebulizing gas (120 psi). The generated products were analyzed by the MS in a positive mode.

## Simulation details

We conducted the molecular dynamics (MD) simulation and density functional theory (DFT) calculation for this study. To obtain the equilibrium structures of toluene and benzaldehyde molecules at the oil–water interface, the following step was processed. The water molecules were packed in the box of 45.975 × 45.975 × 45.975 Å$^3$ (i.e.,

3239 molecules), which was relaxed by the MD simulation with the NPT (i.e., isothermal-isobaric) ensemble for 1 ns at 25 °C and 1 atm. The final cell parameter of the relaxed box was 46.017 × 46.017 × 46.017 Å$^3$. To accommodate 4 toluene or 4 benzaldehyde molecules on the water surface, z-axis was expanded by ~2–3 Å. Subsequently, the 46.017 × 46.017 × 92.050 Å$^3$ box for the oil system was modeled by packing the hexadecane molecules (i.e., 394 molecules) with the density of 0.772 g/cm$^3$ to closely match the experimentally observed concentration of 0.01 M. This system was relaxed by the MD simulation with the NVT (i.e., isothermal) ensemble for 1 ns at 25 °C. Finally, the water and oil model systems were combined and the MD simulation with the NVT ensemble for 5 ns at 25 °C was performed (Supplementary Figs. 22 and 23). The time step was set to 1 fs. For isothermal and isobaric states, the Nose thermostat and Berendsen barostat were employed, respectively. The electrostatic and van der Waals interactions were calculated using the Particle–Particle Particle-Mesh (PPPM) and atom-based summation method, respectively. All systems were described by COMPASS III forcefield in Materials Studio 2023[62,63].

To examine the reaction pathway, the DFT calculation was performed using DMol$^3$ program[64,65]. For H-abstraction and OH-addition reactions, the equilibrated MD configuration was selected to set up the interface environment, which contained toluene (e.a. 1), water (e.a. 7), and oil molecules (e.a. 1). Note that water and oil molecules are selected within 2.5 Å around toluene molecule. The OH· with 3 water molecules (i.e., stabilizing the radical) were located around target reaction sites. In case of interfacial benzaldehyde model, the equilibrated MD configuration was also selected to set up the interface environment, which involved benzaldehyde (e.a. 1), water (e.a. 16), and oil (e.a. 1) molecules. Similarly, the solvent molecules were selected within 2.5 Å around benzaldehyde molecule. For the H-abstraction reaction, the water molecule around the target reaction site was replaced with OH·. For the DFT calculation, the electron exchange-correlation energy was calculated with the generalized gradient approximation and the Perdew–Burke–Ernzerhof (GGA-PBE) functional[66]. The effective core potential was used for core treatment with the basis set of DNP 4.4 level. The convergence criteria for energy, force, and displacement were set to $1.0 \times 10^{-5}$ Ha, 0.002 Ha/Å, and 0.005 Å, respectively. To include the dispersion correction of the van der Waals effect, the Tkatchenko–Scheffler scheme was used[67]. The conductor-like screening model (COSMO) was applied using the dielectric constant of water solvent ($\varepsilon = 78.54$)[68]. To capture the transition states, the linear synchronous transit (LST) and quadratic synchronous transit (QST) methods were applied until satisfying the convergence criteria of the RMS force, which was set to 0.003 Ha/Å[69,70].

## Data availability

All data are presented in the paper and the Supplementary Information. Any additional information can be obtained from corresponding authors upon request. In addition, an excel file is supplied including the atomic coordinate of optimized structure of MD and DFT calculations. Source data are provided with this paper.

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

## Acknowledgements

This research was supported by the Basic Science Research Program through the National Research Foundation of Korea (RS-2023-00276535, S.Q.C.) and the KAIST Institute for the Nanocentury (S.Q.C.). We acknowledge the financial support provided by the National Research Foundation of Korea (NRF-RS-2023-00257666, S.K.K.) and computational resources from KISTI (KSC-2022-CRE-0094, S.K.K.) and UNIST-HPC (S.K.K.).

## Author contributions

K.L. conducted all experiments and wrote the manuscript with contributions from all authors. Y.C. and S.K.K. carried out MD simulations. J.C.K. and S.K.K. performed DFT calculations. C.C. and J.K.L. conducted microdroplet mass spectrometry analysis. J.K. and S.L. carried out SEC analysis. S.Q.C. wrote the paper and supervised the project.

## Competing interests

The authors declare no competing interests.
