## [Peer Review File · Nature Communications]

Catalyst-free selective oxidation of C(sp³)-H bonds in toluene on-waterReviewers' Comments:

Reviewer #1:

Remarks to the Author:

The authors reported the highly selective oxidation of toluene at the water-oil interface of microdroplets, where spontaneously generated hydroxyl radicals at the interface oxidize the C(sp³)-H bonds of toluene, resulting in the formation of benzaldehyde. This work is inspiring as a metal-free catalytic method for the selective oxidation of toluene, especially for industrial fields. However, more details are required to comprehensively explain the chemical nature of the toluene reaction at the water-oil interface. The manuscript requires further revisions to meet the standards for publication in Nature Communications.

1. Line 43. The statement in the introduction regarding the scarcity of works inducing spontaneous redox reactions through on-water chemistry is inaccurate. Water microdroplets represent a highly representative model for studying on-water chemistry. Numerous works based on microdroplets have been published on hydroxyl radical/electron-induced spontaneous redox reactions. Some of these works also focus on C-H bond activation, presenting challenges similar to the selective activation of C(sp³)-H bonds in toluene; oxidation of small molecules (Angew. Chem. Int. Ed. 2022, e202207587); the formation of benzyl anions in toluene and their nucleophilic addition to CO₂ (J. Am. Chem. Soc. 2023, 145, 7724); the C(sp³)-N coupling between toluene and secondary amines (J. Am. Chem. Soc. 2022, 144, 19709); the oxidation of I⁻ (Chem Comm. 2022, 58, 12447); the activation of C(sp³)-H bonds in methane (J. Am. Chem. Soc. 2023, 145, 27198); and a review (JACS Au 2023, 3, 1563-1571). I understand that the authors aim to emphasize the high selectivity of toluene oxidation at the water-oil interface, but such a statement needs to be articulated in an appropriate manner.

2. Line 72. There are existing articles on the generation of hydrogen peroxide at the interface of water with other hydrophobic substances, not solely focusing on air-water interfaces: J. Am. Chem. Soc. 2023, 145, 21538; Nat. Commun. 2022, 13, 130; Angew. Chem. Int. Ed. 2023, 62, e202300604.

3. There are many shortcomings in the DFT calculations. In calculating energy changes in organic systems, hybrid functionals is significantly more accurate than pure functionals. The authors did not provide information about the basis set in the Simulation Details. The authors also did not mention whether dispersion correction was employed, which is crucial for studying hydrogen bonds or weak interactions. Even if the authors employed dispersion correction, to my knowledge, DMol3 program seems to support only DFT-D2 correction. More widely used, advanced, and accurate corrections such as D3 or even D4 would be preferable. Furthermore, the use of COSMO as an implicit solvent model is not appropriate. The COSMO model simplified the calculation of surface charges on the cavity surface around the solute molecule, which makes it one of the coarsest approximations among implicit solvent models. Even its extension (COSMO-RS) did not exhibit satisfactory predictive reliability. IEFPCM and SMD model are much more recommended.

4. Supplementary Fig. 3. It is interesting that the concentration of H₂O₂ decreases with the shortening of the carbon chain of the organic solvent in the oil phase. I am looking forward to a convincing explanation for this phenomenon from the authors. Besides, does it imply the need to consider the contribution of nonpolar components in the implicit solvent model when constructing the water-oil interface model for DFT calculations?

5. In a previous study, Zare et al. confirmed that contact electrification at the water-oil interface can generate hydrogen radicals, and their studied system was also water-hexadecane emulsion (J. Am. Chem. Soc. 2023, 145, 21538). They also observed the formation of shorter alkane radical cations, which are oxidation products of hexadecane. Could these minor products potentially have an impact on this study?

Reviewer #2:

Remarks to the Author:

In this manuscript, Kwak, Choi and co-workers reported a catalyst-free oxidation of toluene and other aromatics in "on-water" catalysis system, showing a high selectivity on C(sp³)-H oxidation. The hydroxyl radical spontaneously generated at water/oil interface is believed to be responsible for the initial C(sp³)-H activation and the unexpected selectivity. While this work highlights interesting and significant features of water/oil interfaces, there are several concerns and questions that should be addressed before considering its acceptance.

Major concerns:

1. Effect of sonochemistry. The oxidation of toluene in water through sonochemistry can be traced back to 1990s. Addressing the potential effect of ultrasound in this study would significantly enhance its novelty. However, the authors claim that "The absence of toluene oxidation in sonicated bulk solutions (Table 1, entry 9) supports that the generation of OH· at the oil-water interfaces assisted by ultrasound energy is the determinant factor for toluene oxidation" (lines 100 to 101). Is this conclusion contradictory to the actual findings?

Additionally, it may not be conclusive enough to entirely rule out the influence of sonochemistry, because only the oil phase was sonicated. The absence of water in this case raises questions, considering water's role as a reactant in sonochemistry to generate H atom and OH radical. Additional experiments are suggested, such as performing sonication of water/toluene mixture, or monitoring toluene oxidation after emulsion formation without sonication.

2. Effect of O₂. This is a significant consideration given the authors claimed this is spontaneous oxidation. In all experimental scenarios, either O₂ or air was used, and the optimal condition was under 1 atm oxygen. Moreover, when changing air to O₂, the conversion increased from 12% to 48%, aligning with the proportional increase of O₂ content in air (see supplementary Table 3). It would be valuable to explore conversion in the absence of O₂. Additionally, the dissolved O₂ in both water and oil phase should be considered.

3. Discussion on H₂O₂. As mentioned in lines 48 to 49, "Hydroxyl radicals (OH·) were spontaneously generated at the oil-water interface and activated the C(sp³)-H bonds in toluene to create benzaldehyde.", However, <0.01% conversion when a substantial amount of H₂O₂ was added (entry 10, Table 1) raises the concern of real mechanism/oxidant responsible for this oxidation. Evidence is needed to support that OH radical can oxidize the toluene. This concern is related to some confusing results, like the generation of near 1 mM H₂O₂ even when the interface is blocked with the surfactant (Supplementary Fig. 14a) and constant increase in H₂O₂ concentration despite the addition of substrates, such as toluene (Supplementary Fig. 3).

4. Significant statement in the introduction. The authors claimed that "Few on-water studies have utilized aqueous interfaces to regulate chemo-selectivity of the final products". However, many reviews and studies already show that both regioselectivity and even stereoselectivity could be regulated by on-water catalysis. Additionally, "to the best of our knowledge, no study has demonstrated the ability of on-water chemistry to induce spontaneous redox reactions ...", while the ref 15 in this manuscript describes the H₂O₂ generation and OH radical-initiated polymerization at water-oil interface. It is highly recommended to restate the significance of this work.

Minor points:

1. In the introduction, lines 31 to 32, "Those unique chemistry occurring at the water-hydrophobe interfaces, which cannot be observed in bulk water". A more specific explanation on the "water-hydrophobe interface" should be given, like water-solid or water-air interface, to emphasize the novelty of this work – water-oil interface.

2. Fig 2c and 2d, what do the error bars mean, and why is the error bar in the w/o =1:1 case smallest while its droplet size distribution is widest among the three emulsion system systems?

3. Lines 121 to 122, "we observed the stable C₇H₇⁺ (m/z 91.0549), generated by the loss of one electron from the benzyl radical¹³, via microdroplet mass spectrometry operating in positive mode (Supplementary Fig. 8)." The observation of C₇H₇⁺ in microdroplets (with air/water interface) does

not necessarily serve as conclusive evidence supporting benzyl radical as the intermediate. As mentioned in the introduction, microdroplet chemistry and on-water chemistry represent different systems.

4. Line 129, "in our system, however, the formation of benzyl radical and further oxidized product benzaldehyde was predominantly observed". What is the evidence for the observation of benzyl radical?

5. Line 156, "in our system, 10 M toluene (100% v/v) still showed a selectivity higher than 90% for C(sp³)-H bond activation" To what result does this conclusion refer?

6. Line 178, "Since it is known that the OH· attached to the benzene prefers to withdraw the electrons from the toluene,[46] the OH addition reaction becomes more challenging in the toluene interacting with water." Is OH· attached to the "benzene" or "toluene"? Additionally, ref 46 does not appear to support this argument, as the major point in ref 46 is that OH radical "preferred to withdraw the electron from the N atom that was conjugated to the benzene ring molecules."

7. Fig 4, y axis "H-abstraction/OH-addition activation ratio". An explanation of how this value was calculated should be provided.

8. Quantification of H₂O₂. The calibration curve of H₂O₂ measurement ranges from 0 to 500 μM (Supplementary Fig. 2.), while in several cases, more than 1 mM H₂O₂ was measured (Supplementary Fig. 3a). An explanation should be provided.

We express our gratitude for the reviewers' time and effort in offering insightful feedback regarding potential areas of improvement for our work. The following is a detailed, point-by-point response to the valuable comments and queries.

The reviewers' comments are delineated in **bold**, while the revised texts are highlighted.

Reviewer #1

The authors reported the highly selective oxidation of toluene at the water-oil interface of microdroplets, where spontaneously generated hydroxyl radicals at the interface oxidize the C(sp³)-H bonds of toluene, resulting in the formation of benzaldehyde. This work is inspiring as a metal-free catalytic method for the selective oxidation of toluene, especially for industrial fields. However, more details are required to comprehensively explain the chemical nature of the toluene reaction at the water-oil interface. The manuscript requires further revisions to meet the standards for publication in Nature Communications.

Response. We appreciate the reviewer for the thoughtful comments pertaining to our work. In consideration of the reviewer's suggestion, we have revised our work in order to enhance its clarity. Attached below are detailed responses:

1. Line 43. The statement in the introduction regarding the scarcity of works inducing spontaneous redox reactions through on-water chemistry is inaccurate. Water microdroplets represent a highly representative model for studying on-water chemistry. Numerous works based on microdroplets have been published on hydroxyl radical/electron-induced spontaneous redox reactions. Some of these works also focus on C-H bond activation, presenting challenges similar to the selective activation of C(sp³)-H bonds in toluene; oxidation of small molecules (Angew. Chem. Int. Ed. 2022, e202207587); the formation of benzyl anions in toluene and their nucleophilic addition to CO₂ (J. Am. Chem. Soc. 2023, 145, 7724); the C(sp³)-N coupling between toluene and secondary amines (J. Am. Chem. Soc. 2022, 144, 19709); the oxidation of I⁻ (Chem Comm. 2022, 58, 12447); the activation of C(sp³)-H bonds in methane (J. Am. Chem. Soc. 2023, 145, 27198); and a review (JACS Au 2023, 3, 1563-1571). I understand that the authors aim to emphasize the high selectivity of toluene oxidation at the water-oil interface, but such a statement needs to be articulated in an appropriate manner.

Response. We appreciate the reviewer for the valuable comment. Our objective was to highlight emphasis to the limited number of works that simultaneously meet the following three criteria: 1) spontaneous redox reactions; 2) activation of C(sp³)-H bonds; and 3) extremely high selectivity and yield. However, our intention to emphasize our work could be misrepresented by our statement. After conducting a comprehensive review of the references, we have revised the statement on **Page 2** in the revised manuscript based on the reviewer's comment. Furthermore, references that illustrated the spontaneous redox reaction via on-water chemistry were incorporated on **Page 1**.

Angew. Chem. Int. Ed. 2022, e202207587 - Ref. #33 in the revised manuscript

J. Am. Chem. Soc. 2023, 145, 7724 - Ref. #12 in the revised manuscript

J. Am. Chem. Soc. 2022, 144, 19709 - Ref. #17 in the revised manuscript

Chem. Comm. 2022, 58, 12447 - Ref. #13 in the revised manuscript

J. Am. Chem. Soc. 2023, 145, 27198 - Ref. #14 in the revised manuscript

JACS Au 2023, 3, 1563-1571 - Ref. #15 in the revised manuscript

Revised Main Text and added References

(Page 2) Despite the power of the water-hydrophobe interfaces to induce unique phenomena, however, ~~Furthermore, to the best of our knowledge, no few studies have demonstrated the ability-potential of on-water chemistry to induce spontaneous redox reactions and to prompt traditionally challenging reactions, such as the activation of C(sp³)-H bonds with high selectivity-activate C(sp³)-H bonds¹⁷. Furthermore, it is particularly rare for C(sp³)-H bonds to be activated in highly selective manner, without the need for a catalyst, and with a significant yield.~~

(Page 1) Those unique chemistry occurring at the ~~interface of sprayed water microdroplets water hydrophobe interfaces~~, which cannot be observed in bulk water, include the marked acceleration of chemical reaction rates^{8,9}, spontaneous redox reactions¹⁰⁻¹⁵, C-N bond formation^{16,17}, amination¹⁸, and carboxylation¹¹, ~~and even polymerization⁴⁵.~~

12. Meng, Y., Gnanamani, E. & Zare, R. N. One-Step formation of pharmaceuticals having a phenylacetic acid core using water microdroplets. *J. Am. Chem. Soc.* **145**, 7724-7728 (2023).
13. Xing, D. *et al.* Spontaneous oxidation of I⁻ in water microdroplets and its atmospheric implications. *Chem. Commun.* **58**, 12447-12450 (2022).
14. Song, X. Basheer, C. & Zare, R. N. Water microdroplets-initiated methane oxidation. *J. Am. Chem. Soc.* **145**, 27198-27204 (2023).
15. Jin, S. *et al.* The spontaneous electron-mediated redox processes on sprayed water microdroplets *JACS Au* **3**, 1563-1571 (2023).
17. Meng, Y., Gnanamani, E. & Zare, R. N. Direct C(sp³)-N bond formation between toluene and amine in water microdroplets. *J. Am. Chem. Soc.* **144**, 19709–19713 (2022).
33. Xing, D. *et al.* Capture of hydroxyl radicals by hydronium cations in water microdroplets. *Angew. Chem. Int. Ed.* **61**, e202207587 (2022).

2. Line 72. There are existing articles on the generation of hydrogen peroxide at the interface of water with other hydrophobic substances, not solely focusing on air-water interfaces: J. Am. Chem. Soc. 2023, 145, 21538; Nat. Commun. 2022, 13, 130; Angew. Chem. Int. Ed. 2023, 62, e202300604.

Response. We appreciate the reviewer's helpful comments. We have mistakenly stated the sentence on **Page 4** (line 72). The statement that prior research concentrated predominantly on sprayed aqueous microdroplets was eliminated. Additionally, references demonstrating the production of OH· at the interfaces of water and hydrophobic substances were also included.

J. Am. Chem. Soc. 2023, **145**, 21538 - Ref. #34 in the revised manuscript

Nat. Commun. 2022, **13**, 130 - Ref. #36 in the revised manuscript

Angew. Chem. Int. Ed. 2023, **62**, e202300604 - Ref. #35 in the revised manuscript

Revised Main Text and added References

(Page 4) This indicates that, as previously observed,^{6,10,11,16-18,32-36} the spontaneous generation of OH· is a general phenomenon of water-hydrophobe interfaces, ~~while previous studies primarily focused on sprayed aqueous microdroplets^{6,10-14,28,29}.~~

34. Chen, X. *et al.* Hydrocarbon degradation by contact with anoxic water microdroplets. *J. Am. Chem. Soc.* **145**, 21538-21545 (2023).

35. Zhao, J. *et al.* Contact-electro-catalysis for direct synthesis of H₂O₂ under ambient conditions. *Angew. Chem. Int. Ed.* **62**, e202300604 (2023).
36. Wang, Z. *et al.* Contact-electro-catalysis for the degradation of organic pollutants using pristine dielectric powders. *Nat. Commun.* **13**, 130 (2022).

3. There are many shortcomings in the DFT calculations. In calculating energy changes in organic systems, hybrid functionals is significantly more accurate than pure functionals. The authors did not provide information about the basis set in the Simulation Details. The authors also did not mention whether dispersion correction was employed, which is crucial for studying hydrogen bonds or weak interactions. Even if the authors employed dispersion correction, to my knowledge, DMol3 program seems to support only DFT-D2 correction. More widely used, advanced, and accurate corrections such as D3 or even D4 would be preferable. Furthermore, the use of COSMO as an implicit solvent model is not appropriate. The COSMO model simplified the calculation of surface charges on the cavity surface around the solute molecule, which makes it one of the coarsest approximations among implicit solvent models. Even its extension (COSMO-RS) did not exhibit satisfactory predictive reliability. IEFPCM and SMD model are much more recommended.

Response. We thank the reviewer's valuable comments. We employed the Perdew-Burke-Ernzerhof (PBE) functional, the Double Numerical basis with polarization (DNP) 4.4 basis set with the all-electron relativistic core treatment, and Tkatchenko–Scheffler (TS) dispersion correction method (*Phys. Rev. Lett.*, 1996, **77**, 3865 - Ref. #64 in the revised manuscript; *Phys. Rev. Lett.*, 2009, **102**, 073005 - Ref. #65 in the revised manuscript). Although the theoretical methods used in this work might not have satisfactory accuracy, we would like to mention that the trend and value of the heat of reaction and activation energy shown in **Fig. 3c** were consistent with those reported in the previous study (*Phys. Chem. Chem. Phys.*, 2020, **22**, 22279 - Ref. #46 in the revised manuscript). More importantly, the energy of the reaction mechanisms was investigated using the calculation setting recommended by the reviewers. Note that main parts of the OH addition and H-abstraction reactions were focused in this calculation applied with DFT calculation setting with a hybrid functional B3LYP (Becke's three-parameter hybrid functional using the LYP correlation functional) at 6-311G+ (d, p) and the DFT-D3 correction by using the Gaussian 16 program (*J. Comput. Chem.*, 2011, **32**, 1456; *J. Chem. Phys.*, 2010, **132**, 154104; *Gaussian 16, Revision C.01; Gaussian, Inc.: Wallingford CT, 2019*). The SMD solvation model was employed to include the implicit solvent effect (*J. Phys. Chem. B*, 2009, **113**, 6378). The activation energy and heat of reaction of the H-abstraction and OH-addition reactions obtained through this set-up showed the same trend with marginal difference in activation energies compared to those calculated in our manuscript (**Fig. R1**). From these results, our calculation settings may not be accurate, yet the predicted results for this case are considered reliable.

Also, to provide clear explanations for the information that was not mentioned in our manuscript, we have revised the Simulation details on **Page 15**.

Figure R1. **a**, Reaction pathways of H-abstraction and OH-addition, and the reaction state configurations of toluene in bulk oil. **b**, OH-addition and **c**, H-abstraction. For a clear view, toluene and water molecules are shown by ball-and-stick model. In the color scheme, C, O, and H atoms are colored in gray, red, and white, respectively. The activation energy and heat of reaction are represented by the number in parenthesis below each snapshot in the unit of kcal/mol. IS, TS, and FS in each reaction mechanism represent initial state, transition state, and final state, respectively.

Main Text

Fig. 3c. Potential energy profile of toluene-OH \cdot reaction in bulk oil

Revised Main Text and added Reference

(Page 15) For the DFT calculation, the electron exchange-correlation energy was calculated with the generalized gradient approximation and the Perdew–Burke–Ernzerhof (GGA-PBE) functional⁶⁴. The effective core potential was used for core treatment with the basis set of DNP 4.4 level. The convergence criteria for energy, force, and displacement were set to 1.0×10^{-5} Ha, 0.002 Ha/Å, and 0.005 Å, respectively. To include the dispersion correction of the van der Waals effect, the Tkatchenko–Scheffler scheme was used⁶⁵. The conductor-like screening model (COSMO) was applied using the dielectric constant of water solvent ($\epsilon = 78.54$)⁶⁶. To capture the transition states, the linear synchronous transit (LST) and quadratic synchronous transit (QST) methods were applied until satisfying the convergence criteria of the RMS force, which was set to 0.003 Ha/Å^{67,68}.

65. Tkatchenko, A. & Scheffler, M. Accurate molecular van der Waals interactions from ground-state electron density and free-atom reference data. *Phys. Rev. Lett.* **102**, 073005 (2009).

4. Supplementary Fig. 3. It is interesting that the concentration of H₂O₂ decreases with the shortening of the carbon chain of the organic solvent in the oil phase. I am looking forward to a convincing explanation for this phenomenon from the authors. Besides, does it imply the need to consider the contribution of nonpolar components in the implicit solvent model when constructing the water-oil interface model for DFT calculations?

Response. The reviewer's constructive remarks are greatly appreciated. While conducting a more comprehensive analysis is required to determine the precise reason for the reduction in H₂O₂ production as the carbon chain length decreases, we put forth the following hypothesis in an effort to elucidate this phenomenon:

- 1) At the water-hydrophobe interfaces, water molecules could develop robust orientations. (*Science*, 2001, **292**, 908 - Ref. #3 in the revised manuscript)
- 2) The alignment of water molecules through the interfaces produces a net dipole, which in turn generates strong electric fields. (*J. Am. Chem. Soc.*, 2008, **130**, 16556 - Ref. #5 in the revised manuscript)
- 3) The inherent strong electric field at the water-hydrophobe interfaces is sufficient to produce OH· from OH⁻. The resulting OH· can be easily recombined to produce H₂O₂. (*Proc. Natl. Acad. Sci. USA*, 2019, **116**, 19294 - Ref. #6 in the revised manuscript)
- 4) However, as the length of the hydrocarbon chain decreases, so does the degree of water orientation at the interfaces. (*Phys. Chem. Chem. Phys.*, 2023, **25**, 5808 - Ref. #37 in the revised manuscript)
- 5) A potential consequence of a shorter carbon chain might be a diminished interfacial area occupied by the strong electric field, which consequently results in a decrease in the rate of H₂O₂ production.

However, further in-depth analysis of the mechanism underlying the carbon chain shortening effect on H₂O₂ production would be required, which we believe should be reserved for another separate study.

If the aforementioned hypothesis is valid, it may be required to consider the contribution of nonpolar components into the implicit solvent model when developing the water-oil interface model for DFT calculations. Nonetheless, this factor may be beyond the scope of our investigation. We conducted DFT calculations to compare enthalpy profiles of H-abstraction and OH-addition in bulk and at the oil-water interfaces. The near-perfect selectivity for benzaldehyde production (initial toluene 0.01 M, **Supplementary Table 5**) was observed irrespective of the chain length of hydrocarbon oils; therefore, it may not be essential to consider nonpolar components in the implicit solvent model in order to elucidate the reaction mechanism of on-water selective toluene oxidation. However, given the potential significance of the effect nonpolar components in the implicit solvent model, a more thorough investigation should be considered in future studies.

We express our apologies for missing this information from the initial draft of the manuscript. According to the reviewer's suggestion, we provided this discussion on **Page 4** in the revised manuscript. Additionally, a reference demonstrating the chain length effect on the orientation of interfacial water was also included.

Supplementary Information

Supplementary Table 5. Impact of chain length of hydrocarbon oils on toluene oxidation

Entry	w:o (v/v)	[Toluene] ₀ (M)	Time (h)	Conv. (%) ^a	Selectivity (%) ^a				
					Benzaldehyde	Benzyl alcohol	Cresol	Bibenzyl	Others
C16	10:1	0.01	10	48.15	> 99	-	-	-	-
C12	10:1	0.01	10	7.42	> 99	-	-	-	-
C8	10:1	0.01	10	0.10	> 99	-	-	-	-

All reactions were conducted under 1 atm oxygen and 25 °C (298 K). ^aConversion and selectivity were analyzed by gas chromatography-mass spectrometry.

Revised Main Text and added References

(Page 4) Other types of oil-water interfaces, including octane, dodecane, benzene, toluene, o-xylene, m-xylene, p-xylene, and 1,2,4-methylbenzene also produced H₂O₂ effectively (Supplementary Fig. 3b, 3c), supporting the high chemical activity of various water-hydrophobe interfaces. The decrease in H₂O₂ production rate induced by the shortening of the carbon chain in the oil phases may be attributed to the reduced interfacial water orientation,³⁷ which consequently leads to a reduction in the potential interfacial area occupied by the strong electric field.

37. Hrahshch, F. & Wilemski G. Effects of molecular size and orientation on the interfacial properties and wetting behavior of water/n-alkane systems: a molecular-dynamics study. *Phys. Chem. Chem. Phys.* **25**, 5808 (2023).

5. In a previous study, Zare et al. confirmed that contact electrification at the water-oil interface can generate hydrogen radicals, and their studied system was also water-hexadecane emulsion (J. Am. Chem. Soc. 2023, 145, 21538). They also observed the formation of shorter alkane radical cations, which are oxidation products of hexadecane. Could these minor products potentially have an impact on this study?

Response. We thank the reviewer elevating the above issue. We analyzed the formation of shorter alkane chains and its potential ramifications on the study we conducted. We commenced by examining the formation of shorter alkane chains, as reported by the Zare group (*J. Am. Chem. Soc.* 2023, 145, 21538 - Ref. #34 in the revised manuscript). In Fig. R2a, we observed the formation of various alkane chains, such as octane (C₈H₁₈), nonane (C₉H₂₀), decane (C₁₀H₂₂), undecane (C₁₁H₂₄), dodecane (C₁₂H₂₆), tridecane (C₁₃H₂₈), tetradecane (C₁₄H₃₀), and pentadecane (C₁₅H₃₂) after 10 h reactions. Nonetheless, the shorter alkane concentration was quite negligible (<< 1%), so it may not have had a significant impact on our study. For a duration of ten hours, both the oil droplet size (Fig. R2b) and the H₂O₂ production rate (Fig. R2c) exhibited a high degree of stability. Furthermore, the conversion of toluene increased linearly, as shown in Fig. 2f. However, given the potential significance of these minor products, their effects should be considered in future studies.

We regret that this information was dropped from the initial draft of the manuscript. According to the reviewer's suggestion, we have revised our manuscript on Page 4 and provided them on Supplementary Fig. 8.

Main Text

Fig. 2f. The impact of reaction time (0.01 M toluene) on toluene conversion and selectivity to benzaldehyde: 1 atm oxygen, 25 °C (298 K), water to oil ratio of 10:1 (v/v).

Revised Main Text and added Reference

(Page 4) We further explored the effect of reaction time on the toluene conversion using emulsified 10:1 (v/v) mixtures of water and hexadecane solutions containing 0.01 M toluene (Fig. 2f, Supplementary Table 1). Conversion of toluene linearly increased over time and all toluene was completely oxidized after 20 hours. While tiny amounts of hexadecane could be degraded to shorter alkane chains (Supplementary Fig. 8a),³⁴ the impact of these minor products appears to be insignificant, as evidenced by the consistent droplet size (Supplementary Fig. 8b), rate of H₂O₂ production (Supplementary Fig. 8c), and toluene oxidation rate (Fig. 2f). Regardless of the reaction time, toluene was consistently oxidized to benzaldehyde, and the generated benzaldehyde remained unchanged even after 30 h reactions. In comparison to previous catalytic studies^{27,28,40,41} (Supplementary Table 2), the oil-water interface exhibited greater benzaldehyde selectivity and the capacity for full conversion under mild reaction conditions.

34. Chen, X. *et al.* Hydrocarbon degradation by contact with anoxic water microdroplets. *J. Am Chem. Soc.* **145**, 21538-21545 (2023).

Revised Supplementary Information

Fig. R2 or Supplementary Fig. 8. Degradation of hexadecane and its effect on the on-water reactions. a, GC-MS analysis of hexadecane degradation. The inset image depicts the overall view of elution profiles. Experimental conditions: 1 atm nitrogen, 25 °C (298 K), water to hexadecane ratio 10:1 (v/v), 10 h reactions. **b**, **c**, Diameter of oil droplets (**b**) and H₂O₂ concentration (**c**) generated by emulsifying 1:10 (v/v) mixtures of water and hexadecane.

Reviewer #2

In this manuscript, Kwak, Choi and co-workers reported a catalyst-free oxidation of toluene and other aromatics in “on-water” catalysis system, showing a high selectivity on C(sp³)-H oxidation. The hydroxyl radical spontaneously generated at water/oil interface is believed to be responsible for the initial C(sp³)-H activation and the unexpected selectivity. While this work highlights interesting and significant features of water/oil interfaces, there are several concerns and questions that should be addressed before considering its acceptance.

Response. We are grateful to the reviewer for the insightful feedback regarding our work. Our work has been revised in response to the reviewer's comments. Responses provided point-by-point are provided below:

Major concerns:

1. Effect of sonochemistry. The oxidation of toluene in water through sonochemistry can be traced back to 1990s. Addressing the potential effect of ultrasound in this study would significantly enhance its novelty. However, the authors claim that “The absence of toluene oxidation in sonicated bulk solutions (Table 1, entry 9) supports that the generation of OH· at the oil-water interfaces assisted by ultrasound energy is the determinant factor for toluene oxidation” (lines 100 to 101). Is this conclusion contradictory to the actual findings? Additionally, it may not be conclusive enough to entirely rule out the influence of sonochemistry, because only the oil phase was sonicated. The absence of water in this case raises questions, considering water's role as a reactant in sonochemistry to generate H atom and OH radical. Additional experiments are suggested, such as performing sonication of water/toluene mixture, or monitoring toluene oxidation after emulsion formation without sonication.

Response. We appreciate the reviewer for the insightful comment and apologize for the imprecise statement. Our intention was to emphasize the significance of the following two criteria: 1) oil-water interfaces and 2) ultrasound energy; thus, we stated "... the generation of OH· at the oil-water interfaces assisted by ultrasound energy is the determinant factor ...". To clarify our statement, we have revised the statement on **Page 4** in the revised manuscript.

In addition, we observed negligible toluene oxidation when aqueous solution was sonicated (**Table 1, entry 10**). Despite the introduction of H₂O₂, which can be dissociated to OH· via ultrasound irradiation (*Ultrason. Sonochem.* 2014, **21**, 1976 - Ref. #47 in the revised manuscript), the conversion of toluene was below 0.01%. This suggests that while sonochemical oxidation of toluene in water is achievable, its influence might be quite insignificant in comparison to the oil-water interfaces produced by ultrasound energy. In order to provide more clarity, we have amended the statement on **Page 4** of the revised manuscript.

Finally, we monitored toluene oxidation and H₂O₂ production after emulsion formation without sonication. To make similar droplet size, solutions of hexadecane and water in a 10:1 (v/v) ratio were first subjected to ultrasonication for a short time. Subsequent to the end of ultrasound, we examined the oxidation of toluene and the production of H₂O₂ (**Fig. R3**). Although the emulsion would undergo demulsification once the ultrasound was deactivated due to the absence of surfactant-like molecules, we hypothesized that the interfacial reactions could be observed prior to complete demulsification. Notably, H₂O₂ production was observed even after ultrasonication was turned off. However, H₂O₂ production ceased within a brief period of time, and there was no additional oxidation of toluene subsequent to the end of ultrasonication. All these above results imply that two requirements, oil-water interfaces and ultrasound energy, might be necessary for the optimal observation of the selective toluene oxidation.

We regret that this information was dropped from the initial draft of the manuscript. According to the

reviewer's suggestion, we have revised our manuscript on **Page 4** and provided them on **Supplementary Fig. 6**.

Revised Main Text

(Page 4) The absence of negligible toluene oxidations in sonicated bulk solutions (Table 1, entries 9 and 10) and oil-in-water emulsions without ultrasound (Supplementary Fig. 6) supports that the generation-existence of OH⁻ at the oil-water interfaces assisted-generated by ultrasound energy is the determinant factor for the toluene oxidation on-water.

Revised Supplementary Information

Fig. R3 or Supplementary Fig. 6. Effect of ultrasound on the on-water reactions. a, Schematic diagram of the experimental procedure. Solutions of hexadecane and water in a 10:1 (v/v) ratio were subjected to ultrasonication for the desired time. Prompt reactions were observed subsequent to the end of ultrasonication. **b**, Concentration of H₂O₂ after 10 min exposure to ultrasound. Dotted line represents the H₂O₂ concentration at 10 min. **c**, Conversion of toluene after 1 h exposure to ultrasound (0.01 M toluene). Dotted line represents the conversion of toluene at 1 h.

2. Effect of O₂. This is a significant consideration given the authors claimed this is spontaneous oxidation. In all experimental scenarios, either O₂ or air was used, and the optimal condition was under 1 atm oxygen. Moreover, when changing air to O₂, the conversion increased from 12% to 48%, aligning with the proportional increase of O₂ content in air (see supplementary Table 3). It would be valuable to explore conversion in the absence of O₂. Additionally, the dissolved O₂ in both water and oil phase should be considered.

Response. We are grateful for the reviewer's helpful feedback. We analyzed toluene oxidation after 20-minute purging process with N₂ gas prior to emulsification. We observed the selective oxidation of toluene (Table R1) and H₂O₂ generation (Fig. R4) when air was replaced with N₂. While oxygen may increase reaction rate and selectivity, our result implies that oxygen may not be an essential reactant for the on-water toluene oxidation.

Toluene oxidation to produce benzaldehyde would be initiated through the formation of a benzyl radical

intermediate via OH·. However, the following reactions can be divided into two distinct reaction pathways: 1) undergo a direct reaction with oxygen to produce benzaldehyde; 2) react with OH· to generate benzyl alcohol; and subsequently, the benzyl alcohol undergoes another reaction with OH· to form benzaldehyde. Enhanced selectivity with dissolved O₂ might be attributed to the acceleration of reaction path 1. Increased OH· concentration with dissolved oxygen in **Fig. R4 (Supplementary Fig. 11)** in the revised Supplementary Information) may assist benzaldehyde production. This inquiry was addressed in the manuscript **Page 5**.

J. Phys. Chem. A 2020, **124**, 5917 - Ref. #45 in the revised manuscript;
Phys. Chem. Chem. Phys. 2020, **22**, 22279 - Ref. #46 in the revised manuscript

Our apologies for the misleading statement. In each experimental scenario, the impact of dissolved O₂ in both the water and oil phases was carefully considered. We stated 1 atm because, prior to emulsification, we purged both water and oil for 20 minutes with air, O₂, or N₂. At 25 °C and 1 atm O₂, the solubility of oxygen in pure water is approximately 1.22×10⁻³ mol/L, and that in hexadecane oil is about 9.94×10⁻³ mol/L.

We regret the omission of this information from the initial draft of the manuscript. According to the reviewer's suggestion, we have provided them on **Supplementary Table 3** and **Supplementary Fig.11**.

Main Text

(**Page 5**) Reduced conversion and selectivity with decreasing dissolved oxygen concentration (**Supplementary Table 3**) indicate that the produced benzyl radicals seem to primarily interact with oxygen molecules and are directly oxidized to benzaldehyde⁴⁵, as opposed to forming benzyl alcohol by OH·^{45,46}. Increased OH· concentration with dissolved oxygen (**Supplementary Fig. 11**) may assist benzaldehyde production.

Revised Supplementary Information

Table R1 or Supplementary Table 3. Influence of dissolved oxygen concentration on toluene oxidation

Entry	P _{gas} (1 bar)	w:o (v/v)	[Toluene] o (M)	Time (h)	Conv. (%) ^{ed}	Selectivity (%) ^{ed}				
						Benzaldehyde	Benzyl alcohol	Cresol	Bibenzyl	Others
1 ^a	O ₂	10:1	10.00	10	2.80	24.68	2.71	4.19	25.18	43.24
2 ^b	Air	10:1	10.00	10	1.74	20.76	2.90	4.36	27.00	44.98
3 ^c	N ₂	10:1	10.00	10	1.82	21.16	3.76	5.29	25.39	44.44
3 ⁴ ^a	O ₂	10:1	0.01	10	48.15	> 99	-	-	-	-
4 ⁵ ^b	Air	10:1	0.01	10	11.90	91.74	8.26	-	-	-
6 ^c	N ₂	10:1	0.01	10	9.24	86.00	14.00	-	-	-

^aPurged for 20 min with oxygen prior to emulsification. ^bPurged for 20 min with air before emulsification. ^cPurged for 20 min with nitrogen before emulsification. ^{ed}Conversion and selectivity were analyzed by gas chromatography-mass spectrometry.

Fig. R4 or Supplementary Fig. 11. Influence of dissolved oxygen on H₂O₂ generation. Both water and oil were exposed to a 20-minute purging process with air or O₂ or N₂ gas prior to emulsification.

3. Discussion on H₂O₂. As mentioned in lines 48 to 49, “Hydroxyl radicals (OH·) were spontaneously generated at the oil-water interface and activated the C(sp³)-H bonds in toluene to create benzaldehyde.”, However, <0.01% conversion when a substantial amount of H₂O₂ was added (entry 10, Table 1) raises the concern of real mechanism/oxidant responsible for this oxidation. Evidence is needed to support that OH radical can oxidize the toluene. This concern is related to some confusing results, like the generation of near 1 mM H₂O₂ even when the interface is blocked with the surfactant (Supplementary Fig. 14a) and constant increase in H₂O₂ concentration despite the addition of substrates, such as toluene (Supplementary Fig. 3).

Response. Thank you for the reviewer’s thoughtful comment and apologize for the ambiguous statement. Perhaps the initial remark could be interpreted as two inquiries: 1) Describe an oxidizing agent (H₂O₂ or OH·). 2) What is the reason behind the relatively high conversion (approximately 50%) observed during on-water toluene oxidation, compared to **entry 10 of Table 1** showing a negligible conversion (less than 0.01%) after 10 hours? Below is a list of the responses to each question:

- 1) Our objective was to show, using **entry 10 of Table 1**, that the reaction between toluene and OH· in bulk solutions, not between toluene and H₂O₂. While simple mixing H₂O₂ and toluene do not oxidize toluene, ultrasound irradiation could dissociate H₂O₂ into OH· (*Ultrason. Sonochem.* 2014, **21**, 1976 - Ref. #47 in the revised manuscript) and initiate toluene oxidation. Typically, toluene reaction with OH· can occur either through OH-addition or H-abstraction. In bulk reactions, the OH-addition dominates overall reactions (approximately 90%), while the H-abstraction contributes the remaining 10% (*J. Phys. Chem. A.* 2020, **124**, 5917 - Ref. #45 in the revised manuscript; *Phys. Chem. Chem. Phys.* 2020, **22**, 22279 - Ref. #46 in the revised manuscript; *J. Atmos. Chem.* 1998, **30**, 209 - Ref. #48 in the revised manuscript). OH-addition predominates in the reaction between toluene and OH· in bulk solution, whereas H-abstraction predominates at the oil-water interface (on-water). We hope that **entry 10 of Table 1** will illustrate the reaction between toluene and OH· in bulk solutions.
- 2) Ultrasound irradiation could dissociate H₂O₂ into OH· (*Ultrason. Sonochem.* 2014, **21**, 1976 - Ref. #47 in the revised manuscript) and initiate toluene oxidation. At a frequency of 1142 kHz, the H₂O₂ dissociation rate is approximately 0.01 mM/h (OH· production rate ≈ 0.02 mM/h) (*Appl. Catal. B* 2009, **90**, 380). Since we utilized a 40 kHz ultrasound frequency, the OH· production rate could be significantly reduced from 0.02 mM/h in bulk environment (**entry 10 of Table 1**).

Conversely, the H_2O_2 production rate at the oil-water interfaces is approximately 20 mM/h ($\text{OH}\cdot$ production rate ≈ 40 mM/h) when $w:o = 1:10$ (v/v) and 2 mM/h ($\text{OH}\cdot$ production rate ≈ 4 mM/h) when $w:o = 10:1$ (v/v) in **Supplementary Fig. 3a, 3b**. Moreover, it is anticipated that the concentration of $\text{OH}\cdot$ near the interface will be greater than the measured values, given that $\text{OH}\cdot$ is predominantly produced and undergoes reactions with toluene at the oil-water interfaces. Therefore, on-water systems may exhibit the relatively higher conversion (50%, $w:o = 10:1$ (v/v)) compared to negligible conversion (less than 0.01%) in **entry 10** of **Table 1** after 10 hours.

Although further investigation is required to identify the precise reason of H_2O_2 production even when the interface is blocked by the C12E6 surfactant, we propose the subsequent hypothesis in an attempt to clarify this occurrence:

- 1) The strong electric field at the water-hydrophobe interfaces is sufficient to produce $\text{OH}\cdot$ from OH^- . The resulting $\text{OH}\cdot$ can be easily recombined to produce H_2O_2 . (*Proc. Natl. Acad. Sci. USA*, 2019, **116**, 19294 - Ref. #6 in the revised manuscript)
- 2) The strong electric field ($\vec{E} \approx 10^9$ V/m) comparable to that of water-hydrophobe interfaces ($\vec{E} \approx 10^9$ V/m) was reported for the C12E6 surfactant interfaces. (*J. Am. Chem. Soc.*, 2021, **143**, 15103 - Ref. #55 in the revised manuscript)
- 3) Consequently, a similar rate of H_2O_2 production might be observed in **Supplementary Fig. 14a** (**Supplementary Fig. 16a** in the revised Supplementary Information).

Furthermore, the constant increase in H_2O_2 concentration, even after adding an oil-soluble substrate, could potentially be attributed to the insoluble nature of H_2O_2 (or $\text{OH}\cdot$) and toluene in oil and water, respectively. This implies that not all produced $\text{OH}\cdot$ may undergo a successful reaction with toluene on-water, as the collision between $\text{OH}\cdot$ and toluene at the interface is required to initiate toluene oxidation. The collision probability may be further reduced due to the short lifetime of $\text{OH}\cdot$. An important amount of the generated $\text{OH}\cdot$ may be recombined to form H_2O_2 in water; consequently, despite the addition of substrates like toluene, a constant increase in H_2O_2 might be observed.

However, we believe that a separate study should be devoted to the in-depth examination of the effect of the substrate and surfactant on H_2O_2 production at the water-hydrophobe interfaces.

Supplementary Fig. 16a. a, Impact of interfacial properties on the formation of H₂O₂.

4. Significant statement in the introduction. The authors claimed that “Few on-water studies have utilized aqueous interfaces to regulate chemo-selectivity of the final products”. However, many reviews and studies already show that both regioselectivity and even stereoselectivity could be regulated by on-water catalysis. Additionally, “to the best of our knowledge, no study has demonstrated the ability of on-water chemistry to induce spontaneous redox reactions ...”, while the ref 15 in this manuscript describes the H₂O₂ generation and OH radical-initiated polymerization at water-oil interface. It is highly recommended to restate the significance of this work.

Response. We appreciate the reviewer for the valuable comment. We regret to inform you that the statement on **Page 2** was previously stated incorrectly. The statement that limited on-water studies demonstrated regulated chemoselectivity was eliminated. Moreover, a sentence demonstrating the power of on-water chemistry to regulate regioselectivity, stereoselectivity, and chemoselectivity was included. We have revised the statement on **Page 2** of the revised manuscript in order to provide more clarity.

In addition, we apologize for the imprecise statement “... no study has demonstrated the ability of on-water chemistry to induce spontaneous redox reactions ...”. Our objective was to highlight emphasis to the limited number of works that simultaneously meet the following three criteria: 1) spontaneous redox reactions; 2) activation of C(sp³)-H bonds; and 3) extremely high selectivity and yield. However, our intention to emphasize our work could be misrepresented by our statement. After conducting a comprehensive review of the references, we have revised the statement on **Page 2** in the revised manuscript based on the reviewer's comment.

Revised Main Text

(Page 2) These reactions can be accelerated by several hundred folds due to the stabilized transition states through the hydrogen bonds or the reagent organizations at water-hydrophobe interfaces^{24,25}. An additional capability of on-water chemistry is its potential to regulate regioselectivity, stereoselectivity, and chemoselectivity²⁰⁻²⁶. Despite the power of the water hydrophobe interfaces to induce unique phenomena, however, the applications of the on-water chemistry have been limited mainly to the acceleration of existing reactions in bulk solutions. Few on-water studies have utilized aqueous interfaces to regulate chemoselectivity of the final products²⁶. Despite the power of the water-hydrophobe interfaces to induce unique phenomena, however, Furthermore, to the best of our knowledge, no few studies have demonstrated the ability potential of on-water chemistry to induce spontaneous redox reactions and to prompt traditionally

challenging reactions, such as the activation of C(sp³)-H bonds with high selectivity activate C(sp³)-H bonds¹⁷. Furthermore, it is particularly rare for C(sp³)-H bonds to be activated in highly selective manner, without the need for a catalyst, and with a significant yield.

Minor points:

1. In the introduction, lines 31 to 32, “Those unique 32 chemistry occurring at the water-hydrophobe interfaces, which cannot be observed in bulk water”. A more specific explanation on the “water-hydrophobe interface” should be given, like water-solid or water-air interface, to emphasize the novelty of this work – water-oil interface.

Response. We appreciate the reviewer’s valuable comment. Based on the reviewer's comment, we have revised the statement on **Page 1** in the revised manuscript

Revised Main Text

(**Page 1**) Those unique chemistry occurring at the interface of sprayed water microdroplets water-hydrophobe interfaces, which cannot be observed in bulk water, include the marked acceleration of chemical reaction rates^{8,9}, spontaneous redox reactions¹⁰⁻¹⁵, C-N bond formation^{16,17}, amination¹⁸, and carboxylation¹¹, and even polymerization¹⁵.

2. Fig 2c and 2d, what do the error bars mean, and why is the error bar in the w/o =1:1 case smallest while its droplet size distribution is widest among the three emulsion system systems?

Response. Thank you for the reviewer’s helpful comment. Despite the widest droplet size distribution, the error bar of the S/V ratio may be the smallest. Consider two water S/V ratios: w:o = 1:1 (v/v) and 1:10 (v/v). Due to the position of the droplet radius in the denominator (S/V ratio of sphere = 3/radius), the error bar associated with the S/V ratio of water may be smaller despite the larger error bar associated with the diameter of the water droplet. Please refer to the example in **Table R2** below.

Table R2. Example of the distribution of droplet size and S/V ratio in various emulsion systems.

w:o (v/v)	Average water droplet diameter (d)	S/V ratio of water (3/(d/2))
1:1	1.5 ± 0.5 μm	4.3 ± 1.5 μm ⁻¹
1:10	4.0 ± 1.5 μm	1.7 ± 0.7 μm ⁻¹

3. Lines 121 to 122, “we observed the stable C7H7+ (m/z 91.0549), generated by the loss of one electron from the benzyl radical¹³, via microdroplet mass spectrometry operating in positive mode (Supplementary Fig. 8).” The observation of C7H7+ in microdroplets (with air/water interface) does not necessarily serve as conclusive evidence supporting benzyl radical as the intermediate. As mentioned in the introduction, microdroplet chemistry and on-water chemistry represent different systems.

Response. We appreciate for the constructive comments from the reviewer. Our goal in the introduction was to emphasize that current microdroplet chemistry focuses mainly with reactions occurring at the inner water interfaces as opposed to those occurring at the outer water interfaces (on-water). We believe that the field of

microdroplet chemistry involves more than just sprayed microdroplets, which are air-water interfaces. Oil-water interfaces have been previously investigated in the field of microdroplet chemistry (*ACS Cent. Sci.*, 2022, **8**, 1265 - Ref. #29 in the revised manuscript; *J. Phys. Chem. B.*, 2020, **124**, 9938 - Ref. #7 in the revised manuscript; *J. Am. Chem. Soc.*, 2023, **145**, 21538 - Ref. #34 in the revised manuscript; *Angew. Chem. Int. Ed.*, 2023, **62**, e202300604 - Ref. #35 in the revised manuscript; *Nat. Commun.*, 2022, **13**, 130 - Ref. #36 in the revised manuscript). The existence of microdroplet chemistry is not substantially affected by whether oil is emulsified to generate an oil-water interface or water is sprayed to generate an air-water interface, provided that the hydrophobicity of the outer phase of the microdroplet (air or oil) is sufficient to form interfaces. However, an advantage of oil-water interfaces is that outer interface analysis (on-water reaction, oil phase) is far easier than gas phase analysis.

In this regard, our microdroplet mass spectrometry in **Supplementary Fig. 8** (**Supplementary Fig. 10** in revised Supplementary Information) demonstrates the significance of toluene-water interfaces in the production of $C_7H_7^+$ or benzyl radicals. Given the immiscibility of toluene and water, the sprayed microdroplet should have toluene-water interfaces (**Supplementary Fig. 8a** (**Supplementary Fig. 10a** in the revised Supplementary Information)). $C_7H_7^+$ generation was not observed when the toluene-water interfaces were eliminated via spraying with toluene and acetonitrile solvents. Thus, we believe that the preceding experiments demonstrate the significance of toluene-water interfaces in $C_7H_7^+$ or benzyl radical formation.

4. Line 129, “in our system, however, the formation of benzyl radical and further oxidized product benzaldehyde was predominantly observed”. What is the evidence for the observation of benzyl radical?

Response. Thank you for the reviewer’s helpful comment. The evidence for the observation of benzyl radical is the observation of $C_7H_7^+$ in **Supplementary Fig. 8 (Supplementary Fig. 10** in revised Supplementary Information). $C_7H_7^+$ could be generated by the loss of one electron from the benzyl radical (*J. Am. Chem. Soc.*, 2022, 144, 19709 - Ref. #17 in the revised manuscript) via microdroplet mass spectrometry operating in positive mode. This inquiry was addressed in the manuscript on **Page 5**.

Main Text

(Page 5) We observed the stable $C_7H_7^+$ (m/z 91.0549), generated by the loss of one electron from the benzyl radical¹³, via microdroplet mass spectrometry operating in positive mode (**Supplementary Fig. 10**).

5. Line 156, “in our system, 10 M toluene (100% v/v) still showed a selectivity higher than 90% for C(sp³)-H bond activation” To what result does this conclusion refer?

Response. We appreciate the reviewer’s careful review. This conclusion indicates the power of on-water reaction environment to selectively activate C(sp³)-H bonds in toluene. Even under 10 M toluene conditions, our system predominantly and selectively activates C(sp³)-H bonds to induce H-abstraction as opposed to OH-addition (10 M toluene: > 90%, 0.01 M toluene: > 99% for C(sp³)-H bond activation). The decrease in selectivity towards benzaldehyde when using 10M toluene may be attributed to a higher concentration of benzyl radical intermediates. These intermediates might be converted to benzyl alcohol by OH· and to bibenzyl by the recombination of two benzyl radicals.

This is an intriguing result given that the OH-addition predominates overall reactions in bulk environment (approximately 90%), while H-abstraction contributes the remaining 10%. In our system, protrusion of free OH groups from interfacial water destabilizes the transition state of the OH-addition by forming π -hydrogen bonds with toluene, while the H-abstraction remains unchanged to effectively activate C(sp³)-H bonds.

6. Line 178, “Since it is known that the OH· attached to the benzene prefers to withdraw the electrons from the toluene,[46] the OH addition reaction becomes more challenging in the toluene interacting with water.” Is OH· attached to the “benzene” or “toluene”? Additionally, ref 46 does not appear to support this argument, as the major point in ref 46 is that OH radical “preferred to withdraw the electron from the N atom that was conjugated to the benzene ring molecules.”

Response. We thank the reviewer’s valuable comments. The OH-addition reaction was the attachment of the OH radical formed in the water solvent to the ortho-carbon of toluene. The word, benzene, mentioned in the line 178 of our manuscript was used to represent the aromatic portion (i.e., benzene ring) of toluene. In addition, the reference 46 (*Chem. Sci.*, 2023, 14, 2229 - Ref. #54 in the revised manuscript) was chosen, rather than supporting the notion that OH radicals take electrons from toluene, to clarify that OH radicals act as withdrawing groups by abstracting electrons from organic molecules contrary to the generally known electron-donating ability of hydroxyl groups. Also, we checked other reference describing OH radicals as acquiring electrons from other substances (*J. Chem. Soc., Faraday Trans. 1*, 1987, 83, 113-124). However, since the previous studies explicitly showing the electron transfer from toluene to OH radical were lack, the atomic charges of OH radicals and toluene were compared before and after the OH-addition reaction (**Fig. R5**). From this result, it was observed that some electrons from toluene had transferred to the OH radical. To make our explanation clearer,

we revised the sentence in the revised manuscript **Page 7**.

Fig. R5. Optimized structures and atomic charges of hydroxyl radical and toluene (a) before and (b) after the OH addition reaction. For a clear view, toluene and hydroxyl radical molecules are depicted using the ball-and-stick model. The C, O, and H atoms are colored in grey, red, and white, respectively

Revised Main Text

(Page 7) Since OH \cdot prefers electron-rich state for its attachment; it tends to act as an electron-withdrawing group when oxidizing agent is around⁵⁴, it is known that the OH \cdot attached to the benzene prefers to withdraw the electrons from the toluene,⁵⁴ the OH addition reaction becomes more challenging in the toluene interacting with water molecules. As a result, the alteration of the electronic structure driven by the π -hydrogen interaction at the oil-water interface could prevent the OH-addition reaction, allowing the selective formation of benzaldehyde.

7. Fig 4, y axis “H-abstraction/OH-addition activation ratio”. An explanation of how this value was calculated should be provided.

Response. We thank the reviewer for the helpful comment. Phenolic byproducts (such as cresol) are classified as OH-addition, while others are classified as H-abstraction. We regret that this information was dropped from the initial draft of the manuscript. According to the reviewer’s suggestion, we have revised our manuscript on **Page 14**.

Revised Main Text

(Page 14) Phenolic byproducts (including cresol) are categorized as OH-addition contributors in the H-abstraction/OH-addition activation ratio calculation, whereas other substances are categorized as H-abstraction contributors.

8. Quantification of H₂O₂. The calibration curve of H₂O₂ measurement ranges from 0 to 500 μ M (Supplementary Fig. 2.), while in several cases, more than 1 mM H₂O₂ was measured (Supplementary Fig. 3a). An explanation should be provided.

Response. We appreciate the reviewer’s thoughtful comment. We measured H₂O₂ concentrations exceeding 500 μ M subsequent to the dilution of collected water solutions: 1) Dilute water phase by factors of 1/100 or 1/10 after desired sonication time; 2) Measure H₂O₂ concentration by UV-Vis spectroscopy; 3) Adjust the dilution effect for the initial concentration of H₂O₂. This inquiry was addressed in the caption of Supplementary Fig.2.

Caption in Supplementary Information

Supplementary Fig. 2. UV-visible absorption intensity corresponding to the H₂O₂ concentration. Absorption peak at 353 nm results from the oxidation of the I⁻ ion to the I₃⁻ ion by the catalytic activity of ammonium molybdate in the presence of H₂O₂. **a**, UV-visible spectra of aqueous solutions containing different concentrations of H₂O₂. **b**, Linear increase of absorption peak at 353 nm with increasing H₂O₂ concentration. Higher concentrations above 500 μM were measured after dilution.

Reviewers' Comments:

Reviewer #1:

Remarks to the Author:

I feel the authors have addressed all the concerns very well, I suggest publication of this work.

Reviewer #2:

Remarks to the Author:

The authors have addressed most of my concerns in the revised paper; however, I maintain skepticism regarding the identity of the oxidizing agent herein. There are a few reasons:

(1) The supplementary Fig.6 newly included in SI presents an interesting "delay" phenomenon that after a pause in ultrasound, the oxidation of toluene ceases while the production of H₂O₂ continues (notably, at this moment, the emulsion hasn't undergone complete demulsification, allowing for the collision between OH radical and toluene). The measured concentrations of H₂O₂ at the pause and endpoint are approximately 0.4 mM and 0.6 mM, respectively. This suggests that roughly 33% of the generated oxidative species (as proposed in the manuscript to be hydroxyl radicals) remain reactive during ultrasound pauses. However, the high concentration of reactive hydroxyl radicals appears to contradict the observed near-zero increase in toluene conversion if the interfacial hydroxyl radical is the sole oxidant.

(2) The supplementary Fig.11 newly included in SI reveals a notable decrease in H₂O₂ production when purged with N₂ (only 0.1 mM/h) compared to air (2 mM/h). However, the toluene conversions in these two cases are quite similar, at 11.9% and 9.24%, respectively. The substantial difference in H₂O₂ generation but comparable toluene conversion highlights the difficulty in explaining this result only by assuming that hydroxyl radical is the sole oxidant.

(3) The explanation for the negligible conversion in bulk reactions (entry 10 in table 1) is not sufficiently convincing. Although the authors reference a study stating, "At a frequency of 1142 kHz, the H₂O₂ dissociation rate is approximately 0.01 mM/h (OH· production rate ≈ 0.02 mM/h) (Appl. Catal. B 2009, 90, 380)." this source does not provide data on OH radical generation. Instead, it investigates the consumption of phenol oxidized by H₂O₂ under ultrasonics, which differs from the simple dissociation of H₂O₂ to OH radicals. Moreover, the referenced study uses H₂O₂ concentrations ranging from 0.58 g/L to 4.76 g/L, whereas this work employs 35% H₂O₂ (w/w), equivalent to 350 g/L (diluted to 47 g/L). Additionally, the argument that "Since we utilized a 40 kHz ultrasound frequency, the OH· production rate could be significantly reduced from 0.02 mM/h in bulk environment" appears questionable, as the referenced study shows that H₂O₂ consumption at 382 kHz is about 10 times greater than at 1142 kHz.

Taking into account all the points, it appears there isn't sufficient evidence to conclusively assert that OH radical is, or at least, the sole oxidant. A recent review explores alternative species generated at the interface (Angew. Chem. Int. Ed. 2024, 63, e20240011). Is it possible that these mentioned species, such as electric fields, water radical cations, and/or superoxide, could function as oxidants in this system? It would be beneficial to include discussions of these possibilities in the main text.

Nevertheless, I find the paper very intriguing and would recommend publication after addressing some minor revisions.

We express our gratitude for the reviewers' time and effort in offering insightful feedback regarding potential areas of improvement for our work. The following is a detailed, point-by-point response to the valuable comments and queries.

The reviewers' comments are delineated in **bold**, while the revised texts are highlighted.

Reviewer #1

I feel the authors have addressed all the concerns very well, I suggest publication of this work.

Response. We appreciate the reviewer for the careful review of our work.

Reviewer #2

The authors have addressed most of my concerns in the revised paper; however, I maintain skepticism regarding the identity of the oxidizing agent herein. There are a few reasons:

(1) The supplementary Fig.6 newly included in SI presents an interesting “delay” phenomenon that after a pause in ultrasound, the oxidation of toluene ceases while the production of H₂O₂ continues (notably, at this moment, the emulsion hasn't undergone complete demulsification, allowing for the collision between OH radical and toluene). The measured concentrations of H₂O₂ at the pause and endpoint are approximately 0.4 mM and 0.6 mM, respectively. This suggests that roughly 33% of the generated oxidative species (as proposed in the manuscript to be hydroxyl radicals) remain reactive during ultrasound pauses. However, the high concentration of reactive hydroxyl radicals appears to contradict the observed near-zero increase in toluene conversion if the interfacial hydroxyl radical is the sole oxidant.

(2) The supplementary Fig.11 newly included in SI reveals a notable decrease in H₂O₂ production when purged with N₂ (only 0.1 mM/h) compared to air (2 mM/h). However, the toluene conversions in these two cases are quite similar, at 11.9% and 9.24%, respectively. The substantial difference in H₂O₂ generation but comparable toluene conversion highlights the difficulty in explaining this result only by assuming that hydroxyl radical is the sole oxidant.

(3) The explanation for the negligible conversion in bulk reactions (entry 10 in table 1) is not sufficiently convincing. Although the authors reference a study stating, “At a frequency of 1142 kHz, the H₂O₂ dissociation rate is approximately 0.01 mM/h (OH· production rate ≈ 0.02 mM/h) (Appl. Catal. B 2009, 90, 380).” this source does not provide data on OH radical generation. Instead, it investigates the consumption of phenol oxidized by H₂O₂ under ultrasonics, which differs from the simple dissociation of H₂O₂ to OH radicals. Moreover, the referenced study uses H₂O₂ concentrations ranging from 0.58 g/L to 4.76 g/L, whereas this work employs 35% H₂O₂ (w/w), equivalent to 350 g/L (diluted to 47 g/L). Additionally, the argument that “Since we utilized a 40 kHz ultrasound frequency, the OH· production rate could be significantly reduced from 0.02 mM/h in bulk environment” appears questionable, as the referenced study shows that H₂O₂ consumption at 382 kHz is about 10 times greater than at 1142 kHz.

Taking into account all the points, it appears there isn't sufficient evidence to conclusively assert that OH radical is, or at least, the sole oxidant. A recent review explores alternative species generated at the interface (Angew. Chem. Int. Ed. 2024, 63, e20240011). Is it possible that these mentioned species, such as electric fields, water radical cations, and/or superoxide, could function as oxidants in this system? It would be beneficial to include discussions of these possibilities in the main text.

Nevertheless, I find the paper very intriguing and would recommend publication after addressing some minor revisions

Response. We appreciate the reviewer for the valuable comment. We concur that OH· in our system might not be the sole oxidant. Discussions regarding the potential oxidants in this system, including electric fields, water radical cations, and superoxide, were added on **Page 9**. A reference that investigated alternative species produced at the interfaces was also incorporated into the revised manuscript.

J. Am. Chem. Soc. 2022, **144**, 7606-7609 - Ref. #32 in the revised manuscript

Angew. Chem. Int. Ed. 2024, **63**, e20240018 - Ref. #59 in the revised manuscript

Angew. Chem. Int. Ed. 2022, **61**, e202210765 - Ref. #60 in the revised manuscript

Revised Main Text and added Reference

(Page 9) As the sole oxidant, we considered OH· generated spontaneously at the oil-water interfaces. However, recent research has investigated alternative species, such as electric fields⁵⁹, water cations^{59,60}, and superoxide^{32,59}, that are produced at the interfaces. These possibilities might have a positive effect on the selective oxidation of toluene, or they might serve as minor oxidants in this study.

32. Mehrgardi, M. A., Mofidfar, M. & Zare, R. N. Sprayed water microdroplets are able to generate hydrogen peroxide spontaneously. *J. Am. Chem. Soc.* **144**, 7606–7609 (2022).

59. Qiu, L. & Cooks, R. G. Spontaneous oxidation in aqueous microdroplets: water radical cation as primary oxidizing agent. *Angew. Chem. Int. Ed.* **63**, e202400118 (2024).

60. Qiu, L. & Cooks, R. G. Simultaneous and spontaneous oxidation and reduction in microdroplets by the water radical cation/anion pair. *Angew. Chem. Int. Ed.* **61**, e202210765 (2022).